# PerFedSI: A Framework for Personalized Federated Learning with Side Information

## Abstract

With an ever-increasing number of smart edge devices with computation and communication constraints, Federated Learning (FL) is a promising paradigm for learning from distributed devices and their data. Typical approaches to FL aim to learn a single model that simultaneously performs well for all clients. But such an approach may be ineffective when the clients' data distributions are heterogeneous. In these cases, we aim to learn personalized models for each client's data yet still leverage shared information across clients. A critical avenue that may allow for such personalization is the presence of client-specific side information available to each client, such as client embeddings obtained from domain-specific knowledge, pre-trained models, or simply one-hot encodings. In this work, we propose a new FL framework for utilizing a general form of client-specific side information for personalized federated learning. We prove that incorporating side information can improve model performance for simplified multi-task linear regression and matrix completion problems. Further, we validate these results with image classification experiments on Omniglot, CIFAR-10, and CIFAR-100, revealing that proper use of side information can be beneficial for personalization.

## 1 Introduction

Federated learning (FL) is a promising paradigm for learning a powerful model from a large amount of data distributed among edge devices. Practical challenges of FL include, for instance, privacy leakage on each client's data, communication and computation constraints of clients, system-level heterogeneity, and heterogeneously distributed data or task labels across clients [1]. In this paper, we focus on the challenge of data heterogeneity. The standard FL approach is Federated Averaging (FedAvg), which aims to learn a single model that minimizes the average loss across clients [2]. This approach is practical when each client has similar data distributions. However, when the data distributions differ significantly across clients, FedAvg can degrade in performance and even fail to converge [3, 4]. In these cases, learning a single global model may not be reasonable since not all of the data across clients may be relevant for solving every client's task.

*Personalized federated learning* addresses data heterogeneity by aiming to learn models tailored to each client's data. A plethora of personalized federated learning methods have been proposed recently, with techniques ranging from learning local and global models that interact via linear mixing [5] or regularization [6], learning a subset of model parameters locally while sharing the rest globally [7–11], and learning hierarchical statistical models consisting of local, global and intermediate parameters [12]; please see Appendix A for additional related works.

While these techniques have demonstrated notable performance improvements over non-personalized methods such as FedAvg in data heterogeneous settings, they still suffer from a critical drawback: they fail to utilize *client-specific side information*. Often in FL settings, clients can readily access

Submitted to 36th Conference on Neural Information Processing Systems (NeurIPS 2022). Do not distribute.

information about their data distribution that is not in the form of samples from the distribution itself, which we refer to as side information. This information is already stored on-device, so does not require any increase in memory, and may be beneficial to the client for solving their task. For example, one can use knowledge such as user age, location, and browsing history stored in smartphones to enhance relevant next-word prediction; one can leverage speaker-specific information for speaker identification to personalize voice assistants. However, to our knowledge, a framework for utilizing such information in FL has not yet been proposed.

We fill this vacancy in the literature by introducing a framework, PerFedSI, to utilize side information for personalized federated learning. Based on this framework, we provide theoretical and empirical evidence to show how one can effectively leverage various types of side information for personalization in settings with different types of data heterogeneity. Our contributions are:

- **Framework for personalization via side information.** We establish parametric and non-parametric forms for levering side information in FL. Our formulation is generic in that it encompasses a variety of methods for employing side information. Yet, it is also specific enough to provide clear avenues for employing side information for personalization. In particular, our formulation highlights two key routes through which side information can augment personalization: biasing the logits and weighting features by their importance.

- **Provable benefits of side information in federated matrix completion.** We study matrix completion with side information, also known as inductive matrix completion (IMC). We show that our practical FL algorithm (essentially FedAvg [2]) converges linearly to the ground-truth solution in expectation under reasonable assumptions. Furthermore, our result reveals that the stronger the side information, the less communication and samples/client are required for convergence. Here, we leverage side information through feature importance weighting. We also analyze how side information can be personalized via label biasing.

- **Empirical benefits of side information.** We conduct experiments on benchmark image datasets (Omniglot, CIFAR-10, and CIFAR-100) with different types of user-specific side information. Our results reveal that leveraging side information via PerFedSI in two different forms can significantly improve personalized FL performance in each case.

**Notations.** Boldface lowercase (uppercase) denotes vectors (matrices). We let $\odot$ denote the element-wise product, $[M]$ denote $\{1, 2, \ldots, M\}$, and $[T]_0$ denote $\{0\} \cup [T]$. $\mathrm{Unif}(S)$ is the uniform distribution over $S$, and $1_{\mathcal{G}}$ is the indicator variable that has value 1 if the event $\mathcal{G}$ occurs and 0 otherwise.

## 2 PerFedSI

In this section we introduce our framework for **Per**sonalized **Fed**erated Learning with **S**ide **I**nformation, termed PerFedSI. We start by providing a non-parametric formulation for the greatest generality. We then give examples of parametric forms and discuss the proposed algorithm.

Suppose there are $M$ clients indexed by $m = 1, \ldots, M$. Each client has a data distribution $p_m$ over $\mathcal{X} \times \mathcal{Y}$, where $\mathcal{X}$ is the input space and $\mathcal{Y}$ is the output space, and a vector of side information $\mathbf{z}_m \in \mathcal{Z}$. Let $\mathcal{F} : \mathcal{X} \times \mathcal{Z} \to \mathcal{Y}$ be a class of functions that represent possible personalization mechanisms. That is, for each $f \in \mathcal{F}$, $f(\mathbf{x}, \mathbf{z})$ gives the predicted label of the input $\mathbf{x}$ for client with side information $\mathbf{z}$. For ease of notation, we define $f^{\mathbf{z}}(\mathbf{x}) \coloneqq f(\mathbf{x}, \mathbf{z})$ for all $\mathbf{x} \in \mathcal{X}$, $\mathbf{z} \in \mathcal{Z}$, and $f \in \mathcal{F}$. The local population loss of a model $f \in \mathcal{F}$ for client $m$ is given by

$$\mathcal{L}_m(f) \coloneqq \mathbb{E}_{(\mathbf{x}_m, \mathbf{y}_m) \sim p_m}[\ell(f^{\mathbf{z}_m}(\mathbf{x}_m), \mathbf{y}_m)],$$

where $\ell : \mathcal{Y} \times \mathcal{Y} \to \mathbb{R}$ is a loss function that measures the closeness between the prediction and true labels, such as the cross-entropy loss for classification or squared loss for regression. Each client is ultimately interested in finding a model $\hat{f}$ that performs well on its population loss $\mathcal{L}_m$, namely to achieve an excess risk $\mathcal{E}_m(\hat{f}^{\mathbf{z}_m}) \coloneqq \mathcal{L}_m(\hat{f}^{\mathbf{z}_m}) - \inf_{f \in \mathcal{F}} \mathcal{L}_m(f^{\mathbf{z}_m})$ close to zero. To operate the training from finitely-many observations, a client often solves an empirical risk minimization problem. Specifically, suppose the client $m$ accesses a dataset $D_m \coloneqq \{(\mathbf{x}_{m,i}, \mathbf{y}_{m,i})\}_{i=1}^{n_m}$ consisting of $n_m$ i.i.d. samples from $p_m$. It sets the following local training objective: $\min_{f \in \mathcal{F}} \hat{\mathcal{L}}_m(f) \coloneqq \frac{1}{n_m} \sum_{i=1}^{n_m} \ell(f^{\mathbf{z}_m}(\mathbf{x}_{m,i}), \mathbf{y}_{m,i})$. Meanwhile, the server aims to minimize the weighted average of the

training losses across clients, with weights proportional to the number of local samples:

$$\min_{f \in \mathcal{F}} \hat{\mathcal{L}}(f) := \sum_{m=1}^{M} \frac{n_m}{N} \hat{\mathcal{L}}_m(f^{\mathbf{z}_m}), \tag{1}$$

where $N := \sum_{m=1}^{M} n_m$. The side information can provide rich indexing of clients' models, which we will discuss below. Next, we discuss parametric forms in which side information may help clients to learn personalized models that generalize well with few samples.

## 2.1 Parametric Forms

We postulate parameterized models that incorporate side information. We use the notation $\boldsymbol{\theta} \in \mathbb{R}^d$ to represent the vectorized parameters of a model, and $f_{\boldsymbol{\theta}}^{\mathbf{z}}$ to denote a model with parameters $\boldsymbol{\theta}$ and side information $\mathbf{z}$. The parameterized version of (1) is as follows:

$$\min_{\boldsymbol{\theta} \in \mathbb{R}^D} \mathcal{L}(\boldsymbol{\theta}) := \sum_{m=1}^{M} \frac{n_m}{N} \mathcal{L}_m(f_{\boldsymbol{\theta}}^{\mathbf{z}_m}). \tag{2}$$

The above objective may take many forms, depending on the learning model and how side information is incorporated. We will provide two such examples. In recognition that side information may confer different benefits and should be leveraged differently in various settings, we keep these examples generic to encompass broad uses of side information.

**Model 1: Concatenation.** The first method for incorporating side information is to concatenate the side information with intermediate layer inputs. That is, we learn a model of the form:

$$f_{\boldsymbol{\theta}}^{\mathbf{z}}(\mathbf{x}) = H_{\boldsymbol{\theta}_3}([G_{\boldsymbol{\theta}_1}(\mathbf{x}), W_{\boldsymbol{\theta}_2}(\mathbf{z})]) \tag{3}$$

where $\boldsymbol{\theta} = [\boldsymbol{\theta}_1; \boldsymbol{\theta}_2; \boldsymbol{\theta}_3]$, $G_{\boldsymbol{\theta}_1} : \mathcal{X} \to \mathbb{R}^{d_1}$ is an embedding of the input data, $W_{\boldsymbol{\theta}_2} : \mathcal{Z} \to \mathbb{R}^{d_2}$ is an embedding of the side information, and $H_{\boldsymbol{\theta}_3} : \mathbb{R}^{d_1+d_2} \to \mathcal{Y}$ is a network that maps the concatenated embeddings to $\mathcal{Y}$. The (transformed) side information $W_{\boldsymbol{\theta}_2}(\mathbf{z})$ can be concatenated at any point in the network, depending on the setting.

If $H_{\boldsymbol{\theta}_3}$ is a fully-connected NN with a final softmax layer, as is often the case for the final layers of networks trained for classification, then the side information serves as a client-specific *bias*, since the model outputs can be written as $f_{\boldsymbol{\theta}}^{\mathbf{z}}(\mathbf{x}) = \sigma(H_{\boldsymbol{\theta}_3}^{'}(G_{\boldsymbol{\theta}_1}(\mathbf{x})) + H_{\boldsymbol{\theta}_4}^{''}(W_{\boldsymbol{\theta}_2}(\mathbf{z})))$ for some mappings $H_{\boldsymbol{\theta}_3}^{'}$ and $H_{\boldsymbol{\theta}_4}^{''}$, and $\sigma$ being the softmax activation. As we show in Section 4, this use of side information can be especially beneficial in heterogeneous data settings with *label shift*, wherein each client has samples from only a small subset of $\mathcal{Y}$. Then, the side information can up-weight the logits for popular classes for each client, leading to higher personalized accuracy. If $H_{\boldsymbol{\theta}_3}$ is a Convolutional Neural Network (CNN), similar behavior (up to normalization) of the side information serving as a bias also holds. Concatenating the side information may do more than simply biasing the predictions. For example, suppose the side information is concatenated to the inputs of long short-term memory (LSTM) blocks in a particular layer. In that case, it is ultimately used to parameterize a non-linear function of the features.

**Model 2: Element-wise multiplication.** The second generic parametric model that utilizes side information applies the (transformed) side information as a "mask" by element-wise multiplying it with an intermediate layer output. The intuition for this approach is that the user-specific mask selects the important features for that user's task. Formally, we propose to learn models of the form:

$$f_{\boldsymbol{\theta}}^{\mathbf{z}}(\mathbf{x}) = H_{\boldsymbol{\theta}_3}(G_{\boldsymbol{\theta}_1}(\mathbf{x}) \odot W_{\boldsymbol{\theta}_2}(\mathbf{z})). \tag{4}$$

Ideally, the side information embedding $W_{\boldsymbol{\theta}_2}(\mathbf{z})$ up-weights the important features in $G_{\boldsymbol{\theta}_1}$ and down-weights the non-important ones in accordance with each client's data distribution. This approach can be utilized in settings in which there may exist features of the input data that are broadly relevant across clients, but their importance for each client's task varies in a manner revealed by the side information. For instance, if $W_{\boldsymbol{\theta}_2}(\mathbf{z}_m)$ is sparse, the side information removes a large number of features which are irrelevant for client $m$'s task.

**Remark 2.1** (Personalization without local parameters)**.** All parameters are global in the aforementioned parametric forms, meaning they are commonly shared by all clients. We can modify these forms to allow for additional personalization by including local, client-specific parameters as in other personalized federated learning approaches [8, 9, 11, 7, 5]. However, the more local parameters, the

larger sample size per client required to learn them – leading to poor performance in settings with few samples per client. Thus, we leave the parameters as global to highlight that *employing side information can yield personalization without local parameters.*

**Remark 2.2** (Adaptivity to cases with weak side information). In some cases, the side information may not contain useful information. We want models to be robust to these scenarios by not relying critically on the side information. Both Model 1 and Model 2 have this potential since Model 1 can learn a $W_{\boldsymbol{\theta}_2}$ that maps to the vector of zeros, and Model 2 can learn a $W_{\boldsymbol{\theta}_2}$ that maps to the vector of ones to ignore the side information.

**Algorithm.** PerFedSI employs the FedAvg algorithm [2] to solve (2). FedAvg alternates between local updates and aggregation as the server. At communication round $t$, the server sends the current global model $\boldsymbol{\theta}_t$ down to a batch of selected clients $\mathcal{B}_t$. Then, each selected client $m$ executes $\tau$ steps of SGD on its local data starting from $\boldsymbol{\theta}_t$, i.e. it computes $\boldsymbol{\theta}_{t,m,s+1} = \boldsymbol{\theta}_{t,m,s} - \eta \hat{\mathbf{g}}_{t,m,s}(\boldsymbol{\theta}_{t,m,s})$ for $s = 0, \ldots, \tau - 1$ and $\boldsymbol{\theta}_{t,m,0} := \boldsymbol{\theta}_t$, where $\hat{\mathbf{g}}_{t,m,s}(\boldsymbol{\theta}_{t,m,s})$ is an unbiased stochastic gradient of client $m$'s loss evaluated at $\boldsymbol{\theta}_{t,m,s}$. Then, the clients send $\boldsymbol{\theta}_{t,m,\tau}$ back to the server, which computes the next global iterate $\boldsymbol{\theta}_{t+1} = \frac{1}{|\mathcal{B}_t|} \sum_{m \in \mathcal{B}_t} \boldsymbol{\theta}_{t,m,\tau}$. This synchronous procedure repeats until convergence. Importantly, client $m$'s private side information $\mathbf{z}_m$ is never communicated with the server.

# 3    Theoretical Analysis

In this section, we analyze how side information can improve personalized FL performance via an instance of Model 2. We defer analysis of a Model 1 example – multi-task linear regression with personalized biases – to Appendix B in the interest of space. To show the benefit of side information via Model 2, we study a version of the well-known matrix completion problem [13]. In matrix completion, we aim to learn a rank-$r$ matrix $\mathbf{L}_* \in \mathbb{R}^{d \times M}$ from a strict subset of its entries. Often $\mathbf{L}_*$ is a ratings matrix in which column $m$ gives user $m$'s ratings for each item. We denote $r = \text{rank}(\mathbf{L}_*)$ and assume $r \ll \min(d, M)$. In the federated setting, the server aims to learn a model that allows each client to accurately predict its own ratings, while maintaining the privacy of the ratings, as in e.g. private movie recommendation systems. The key that enables this is side information.

We assume there is a matrix $\mathbf{Z} \in \mathbb{R}^{M \times k}$, $k \geq r$, whose $m$-th row is an embedding of client $m$. This embedding is held by client $m$ as side information, and is informative in the sense that the column space of $\mathbf{Z}$ contains the column space of $\mathbf{L}_*$. Thus, we can re-write $\mathbf{L}_* = \mathbf{M}_* \mathbf{Z}^\top$ for a rank-$r$ matrix $\mathbf{M}_* \in \mathbb{R}^{d \times k}$. The server aims to learn $\hat{\mathbf{M}} \approx \mathbf{M}_*$ in order to allow each client to predict its ratings by computing $\hat{\mathbf{M}} \mathbf{z}_m \approx \mathbf{L}_{*,m}$, where $\mathbf{L}_{*,m}$ is the $m$-th column of $\mathbf{L}_*$. To protect the privacy of both the clients' embeddings and their ratings, $\mathbf{Z}$ is not shared with the server, so the server cannot compute $\mathbf{L}_*$ even if it knows $\mathbf{M}_*$. Nevertheless, we can see how side information is beneficial for the learning process despite not being shared with the server. The smaller $k$, the stronger the side information and the fewer parameters the server needs to learn.

Since $\mathbf{M}_*$ is rank-$r$, the server tries to learn two thin matrices $\mathbf{U} \in \mathbb{R}^{d \times r}$ and $\mathbf{V} \in \mathbb{R}^{k \times r}$ such that $\mathbf{U}\mathbf{V}^\top \approx \mathbf{M}_*$. That is, given input $\mathbf{e}_i$ for client $m$, the learning model predicts $\mathbf{e}_i^\top \mathbf{U}\mathbf{V}^\top \mathbf{z}_m$. In this way we can see that the learning model is an instance of (4), with $G_{\boldsymbol{\theta}_1}(\mathbf{e}_i) = \mathbf{U}^\top \mathbf{e}_i$, $W_{\boldsymbol{\theta}_2}(\mathbf{z}_m) = \mathbf{V}^\top \mathbf{z}_m$, and $H(\cdot)$ fixed as the Sum($\cdot$) operation. Moreover, the side information $\mathbf{z}_m$ provides a client-specific weighting of the input features $\mathbf{U}^\top \mathbf{e}_i$. The global loss is:

$$\mathcal{L}(\mathbf{U}, \mathbf{V}) := \frac{1}{M} \sum_{m=1}^{M} \left\{ \mathcal{L}_m(\mathbf{U}, \mathbf{V}) := \sum_{i=1}^{d} (\mathbf{e}_i^\top (\mathbf{U}\mathbf{V}^\top - \mathbf{M}_*) \mathbf{z}_m)^2 \right\} = \frac{1}{2M} \|(\mathbf{U}\mathbf{V}^\top - \mathbf{M}_*)\mathbf{Z}^\top\|_F^2 \quad (5)$$

where $\mathbf{e}_i$ is the $i$-th standard basis vector in $\mathbb{R}^d$. The local updates involve stochastic gradient updates on the local losses $\mathcal{L}_m(\mathbf{U}, \mathbf{V})$ as detailed in Appendix C.3. Although matrix completion with side information (also known as inductive matrix completion) has been well-studied (please see Appendix A for details), to our best knowledge, no work has shown that (5) can be minimized efficiently by FedAvg, which is difficult to analyze because it executes multiple updates on local losses between communication rounds. These local updates can be problematic in data heterogeneous settings because local gradients may drift away from global gradients, causing FedAvg to not solve the global objective [3, 4]. However, this is not an issue for this problem in simulations with Gaussian data (please see Figure C.4), and we show in Theorem 3.3 that as long as the iterates satisfy regularity properties throughout, then the product $\mathbf{U}_t \mathbf{V}_t^\top$ linearly converges in expectation to $\mathbf{M}_*$. The crux is

that as long as the regularity conditions are satisfied, then the *average* local gradient is close to the global gradient, which leads to convergence. Letting $\mathbf{E}_{t,m,s} := \mathbf{U}_{t,m,s}\mathbf{V}_{t,m,s}^\top - \mathbf{M}_*$, $\sigma_{1,*} := \|\mathbf{M}_*\|_2$ and $\sigma_{r,*} := \sigma_r(\mathbf{M}_*)$, where $\sigma_r(\mathbf{M}_*)$ is the $r$-th singular value of $\mathbf{M}_*$, then the event that the iterates satisfy regularity properties on the $s$-th local update of round $t$ is defined as follows.

**Definition 3.1** (Iterates are regular). Define $\mathcal{A}_{0,0} := \{(\mathbf{U}_0, \mathbf{V}_0) : \mathcal{L}(\mathbf{U}_0, \mathbf{V}_0) \leq c_0\sigma_{r,*}^2, \max(\|\mathbf{U}_0\|_2, \|\mathbf{V}_0\|_2) \leq c\sigma_{1,*}\}$ for some constants $c_0, c$. Furthermore, for all $(t, s) \in \{[T]_0 \times [\tau]_0\} \setminus (0, 0)$ and constant $\mu$, define

$$\mathcal{A}_{t,s} := \Bigg\{ \{(\mathbf{U}_{t,m,s}, \mathbf{V}_{t,m,s})\}_{m \in [M]} : \max(\|\mathbf{U}_{t,m,s}\|_2, \|\mathbf{V}_{t,m,s}\|_2) \leq c\sqrt{\sigma_{1,*}},$$

$$\max_{i \in [d]} \|\mathbf{e}_i^\top \mathbf{U}_{t,m,s}\| \leq \sqrt{\frac{\mu r \sigma_{1,*}}{d}}, \quad \max_{i \in [d]} \|\mathbf{e}_i^\top \mathbf{E}_{t,m,s}\mathbf{z}_m\|_2 \leq \sqrt{\frac{\mu}{d}}\|\mathbf{E}_{t,m,s}\mathbf{z}_m\|_2,$$

$$\mathcal{L}_m(\mathbf{U}_{t,m,s}, \mathbf{V}_{t,m,s}) \leq c\min(\mathcal{L}(\mathbf{U}_t, \mathbf{V}_t), c_0\sigma_{r,*}^2) \Bigg\}.$$

We define $\mathcal{G}_{t,s} := (\cap_{t'=1}^{t} \cap_{s'=1}^{\tau} \mathcal{A}_{t',s'}) \bigcap \cap_{s'=0}^{s} \mathcal{A}_{t,s'}$. If $\mathcal{G}_{T-1,\tau}$ holds, then the norms of $\mathbf{U}_{t,m,s}$ and $\mathbf{V}_{t,m,s}$ are balanced, $\mathbf{U}_{t,m,s}$ and the error $\mathbf{U}_{t,m,s}\mathbf{V}_{t,m,s}^\top - \mathbf{M}_*$ are incoherent with respect to the standard basis and the local loss is never more than a constant factor of the most recent global loss for all $t \in [T-1]_0, m \in [M]$, and $s \in [\tau]_0$. Theorem 3.3 bounds $\mathcal{L}(\mathbf{U}_T, \mathbf{V}_T)1_{\mathcal{G}_{T-1,\tau}}$, so it is only meaningful when $\mathcal{G}_{T-1,\tau}$ holds. Next, we assume a reasonable scaling of $\mathbf{Z}$.

**Assumption 3.2** (Scaling, incoherence of side information). There exist constants $1 \leq c_z, \mu_z < \infty$ such that $\frac{M}{c_z}\mathbf{I}_k \preceq \mathbf{Z}^\top\mathbf{Z} \preceq c_z M\mathbf{I}_k$ and $\max_{m \in [M]} \|\mathbf{z}_m\|_2^2 \leq \mu_z k$.

Now we informally state our main result. The formal statement and proof are found in Appendix C.3.

**Theorem 3.3** (Informal). *Suppose $\eta = O(\frac{1}{k^{3/2}r\tau})$ and Assumption 3.2 holds. Then FedAvg run on (5) with a constant number of clients participating per round and fresh samples drawn on each local update converges linearly to the ground-truth matrix in expectation, namely, for a constant $c'$,*

$$\mathbb{E}[\mathcal{L}(\mathbf{U}_T, \mathbf{V}_T)1_{\mathcal{G}_{T-1,\tau}}] \leq (1 - \frac{c'\eta\tau}{d})^{T-1}\mathcal{L}(\mathbf{U}_0, \mathbf{V}_0). \tag{6}$$

**Benefit of side information**. Theorem 3.3 shows that $T = O(dk^{3/2}r\log(1/\epsilon))$ communication rounds are required to achieve $\epsilon$-error in terms of the global population loss (5) in expectation. Since each client makes $O(\tau/M)$ samples per round on average, this implies that $\tilde{O}(dk^{3/2}r/M)$ samples are required per client, so the clients benefit from stronger side information (smaller $k$). Without collaboration, client $m$ would need $d$ observations to learn its ground-truth solution $\mathbf{M}_*\mathbf{z}_m \in \mathbb{R}^d$, so it benefits from participating in FL as long as the side information is sufficiently strong ($k^{3/2} \ll M/r$). Moreover, in the centralized setting without side information, the information-theoretic lower bound on the sample size required to recover $\mathbf{L}_*$ is $\Omega((d+M)r)$ [14], so using side information can improve on this bound when $k^{3/2} \ll M$. A limitation is that our result is for recovery in expectation, but it can be extended to a high-probability guarantee using martingale analysis [15] in future work.

# 4 Experiments

In this section, we experimentally investigate how side information can be leveraged effectively in settings involving various forms of data heterogeneity. Full details are deferred to Appendix D.

**Baselines.** We compare against five baselines in all experiments, none of which use side information: (1) FedAvg [2]; (2) Ditto [16], a method that learns local models subject to regularization penalizing their distance from a global model; (3) SR-PH, i.e., learning a shared representation and personalized 'head,' or last layer of the model, as in [8, 9]; (4) PR-SH, i.e., learning personalized representations and a shared head, as in [7]; (5) Local, i.e., performing only local training without any communication. All methods sample 20% of clients and execute one epoch of SGD locally on each round.

**Omniglot.** We start with the Omniglot dataset [17], which consists of images of 1623 handwritten characters from 50 different languages. To simulate a realistic heterogeneous dataset, we assign images to clients, so each client's images are from a single alphabet. In other words, each client has observations from classes (characters) belonging to only one out of 50 possible alphabets. The model is a four-layer CNN with a final linear layer. For side information, we train an alphabet classifier

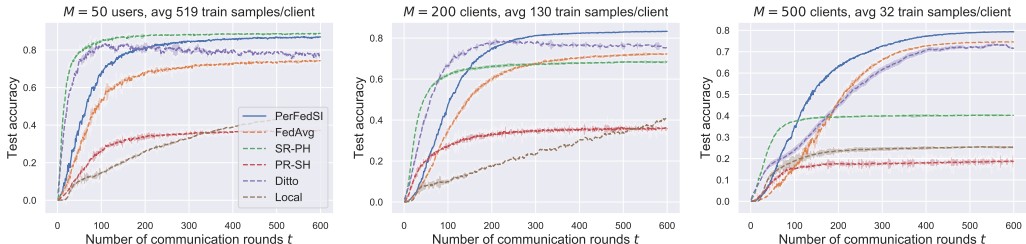

Figure 1: Omniglot test accuracies for varying number of clients (and samples per client).

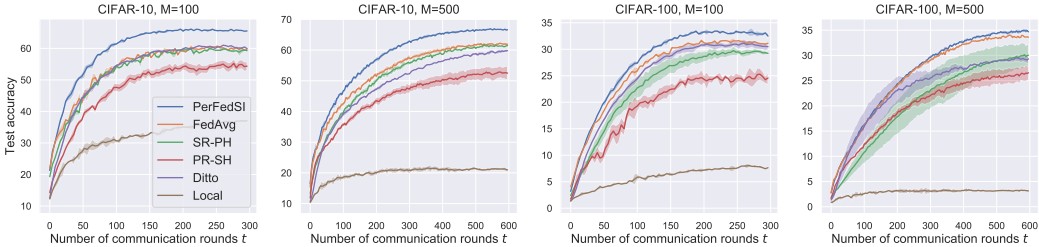

Figure 2: CIFAR-10, CIFAR-100 test accuracies with $M \in \{100, 500\}$ clients and heterogeneity due to affine shifts of the images. Standard deviations over three random trials are shown.

using all training samples across clients. Then, client $m$'s side information is the average embedding output by the alphabet classifier on its training data. Thus, the side information reveals relationships between users, as two clients have similar side information if their samples come from the same or similar alphabets, analogous to client embeddings that may serve as side information in practice. Here, PerFedSI concatenates a two-layer mapping of the side information to the input to the network's final (linear) layer, meaning it takes the form of Model 1, and the side information biases the logits.

Figure 1 plots the test accuracies against communication rounds for all methods, with varying numbers of clients (and hence, training samples per client) in each plot. On the left, there are 50 clients (one client per alphabet) and an average of 519 training samples per client. On the right, there are 500 clients (ten clients per alphabet) and an average of 32 training samples per client. SR-PH performs best when there are many samples per client (left), but PerFedSI achieves the highest test accuracy when there are fewer samples per client (right). In the latter case, local parameters overfit, whereas PerFedSI utilizes side information for personalization without relying on local parameters. A limitation is that public data is required for training the alphabet classifier, however, in practice there is often such a dataset (or embeddings from a pre-trained user identification model) available.

**CIFAR-10, CIFAR-100.** Next, we experiment with CIFAR-10 and CIFAR-100 [18], two image classification datasets with 10 and 100 classes, respectively. Unlike the previous experiment in which the data heterogeneity was due to each client having samples from a small fraction of the total classes, here we realize data heterogeneity by applying different affine shifts to the input data across clients. Specifically, we first partition data i.i.d. among clients, then apply one of four affine shifts consisting of a rotation followed by a shearing operation. These affine shifts represent different camera settings among clients. The side information is a four-dimensional one-hot encoding of the particular shift applied to each client. Again we use a four-layer CNN with a fifth linear layer, but PerFedSI uses an instance of Model 2. In particular, we multiply the side information with the features in the third convolutional layer. Since the side information is a one-hot encoding, some channel outputs are set to zero. Thus, the side information serves as a mask that selects the features relevant to each client. Figure 2 shows that PerFedSI achieves the best test accuracy in all four settings.

**Conclusion.** We have introduced PerFedSI, to our best knowledge, the first framework for utilizing client-specific side information for personalized FL. PerFedSI is general enough to encompass various uses of side information, and we provide theoretical and empirical evidence supporting how particular methods for leveraging side information can improve performance. Future work remains to characterize the benefit of side information in FL from a learning-theoretic standpoint and perform further experiments to obtain a broader picture of when it is useful for personalization.

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

## A  Related Work

**Learning with side information.** Many works have noticed the utility of side information for learning in centralized settings [19]. This line of work has included using side information in the form of word2vec embeddings for natural language processing [20], movie year and overall rating for movie recommendation [21], camera angle and height, caption embeddings, and depth information for computer vision [22–24]. Other works have analyzed how to leverage structural side information for learning Bayesian networks [25–27] and state space side information for reinforcement learning [28]. The two most similar works to our are [29] and [30]. [29] considers that each data point comes with side information in the form of an "index" variable that is correlated with the label of the data point but is insufficient to predict the label on its own. They propose a post-processing procedure that smoothes the predictions based on their index. Our work also leverages side information in the form of indexing, but over clients in FL, not over individual data points in centralized learning. In particular, their smoothing technique would not apply to FL because it would require each client to access the predictions of other clients. [30] proposed a differentially-private algorithm that employs side information for feature importance weighting in distributed settings, including FL. However, the considered side information is global, ideally obtained from a public dataset, whereas we consider client-specific side information that may benefit personalization.

**Personalized FL.** In recent years, there has been a surge of interest in Personalized FL; please see [31] for a detailed summary. The two high-level approaches to Personalized FL are to (i) learn a set of global parameters that can be easily adapted to local datasets and (ii) learn local (device-specific) parameters that can be effectively combined with global parameters (shared across all devices) to yield high-performing personalized models. Approaches of the form (i) start by learning a global model via meta-learning [32, 33] or a general-purpose FL algorithm [2, 4], then fine-tuning this model on each client to obtain personalized models [34, 35]. In contrast, we aim to learn global parameters that effectively leverage side (device-specific) information to give personalized predictions for each client without any fine-tuning needed. Approaches of the form (ii) aim to balance local and global information by either learning local models that are combined with a global model via linear interpolation [5, 36], regularization [6, 16], or hierarchical statistical methods [12], or, they learn a subset of a single model's parameters locally and the rest of the parameters globally [37, 10, 38, 9, 8]. Our work generalizes such approaches since a special case of our framework is the case that the side information is a one-hot vector indicating the index of the device index.

**Inductive matrix completion.** The problem of inductive matrix completion was originally motivated by movie recommendations wherein the goal is to recover a client-movie rating matrix given a subset of its elements and side information about the clients and movies [39]. IMC has also been studied in the context of disease prediction with genetic data as side information, link prediction in networks using features of the nodes as side information, and multi-label learning with features of the inputs as side information [40]. A variety of IMC algorithms have been analyzed, including nuclear norm minimization [40], alternating minimization [39], multi-phase Procrustes flow [41], and Gauss-Newton iteration [42]. Interestingly, simulations show that simple gradient descent and simple variants often perform at least as well as these more sophisticated methods [42]. In this work, we present the first study of whether FedAvg, a simple gradient-based method, can solve this problem while realizing the sample complexity-benefits of using side information.

## B  Model 1 Toy Example: Multi-Task Linear Regression

To demonstrate the advantages of employing side information for personalized FL via Model 1, we study a simplified version of multi-task linear regression with a ground-truth model. Suppose that

data $(\mathbf{x}_m, y_m)$ for the $m$-th client are drawn from the distribution $p_m$ as follows:

$$\mathbf{x}_m \sim p_x, \quad y_m = \langle \boldsymbol{\theta}_*, \mathbf{x}_m \rangle + b_{m,*} + \zeta_m, \tag{7}$$

where $p_x$ has mean zero and identity covariance, $\zeta_m$ is mean-zero random noise, $\boldsymbol{\theta}_* \in \mathbb{R}^d$ encodes the shared information across clients in the form of a ground-truth regressor, and $b_{m,*}$ encodes data heterogeneity in the form of a client-specific bias, analogous to a label shift in classification. Without side information, client $m$ aims to find a model $(\boldsymbol{\theta}, b)$ that achieves small excess risk $\mathcal{E}_m(\boldsymbol{\theta}, b) = \frac{1}{2}\|\boldsymbol{\theta} - \boldsymbol{\theta}_*\|_2^2 + \frac{1}{2}|b - b_{m,*}|^2$. Given $n$ samples $\{(\mathbf{x}_{m,j}, y_{m,j})\}_{j=1}^n$ from each distribution $p_m$, the standard server objective is to minimize the average loss across clients:

$$\min_{\boldsymbol{\theta} \in \mathbb{R}^d, b \in \mathbb{R}} \frac{1}{2Mn} \sum_{m=1}^M \sum_{j=1}^n \left\{ (\langle \boldsymbol{\theta}, \mathbf{x}_{m,j} \rangle + b - y_{m,j})^2 \equiv (\langle \boldsymbol{\theta} - \boldsymbol{\theta}_*, \mathbf{x}_{m,j} \rangle + b - b_m - \zeta_{m,j})^2 \right\}. \tag{8}$$

We can show that as the sample size $n$ goes to infinity, the solution to (8) approaches $(\boldsymbol{\theta}_*, \bar{b}_* :=$ $\frac{1}{M} \sum_{m=1}^M b_{m,*})$. Thus, even in the ideal setting of each client having infinite samples, solving (8) results in each client having excess risk $(b_{m,*} - \bar{b}_*)^2$, which, on average over $m$, grows with the degree of data heterogeneity.

Now suppose each client has side information in the form of client embedding $\mathbf{z}_m \in \mathbb{R}^k$ that encodes some information that distinguishes their data distribution from other clients' data distributions, which in this case corresponds to information about $b_m$. The server aims to learn a model the form (3), where $G_{\boldsymbol{\theta}_1}(\mathbf{x}) = \boldsymbol{\theta}_1^\top \mathbf{x}$, $W_{\boldsymbol{\theta}_2}(\mathbf{z}) = \boldsymbol{\theta}_2^\top \mathbf{z}$, and $H(\cdot)$ is fixed as the identity mapping, by solving

$$\min_{\boldsymbol{\theta}_1 \in \mathbb{R}^d, \boldsymbol{\theta}_2 \in \mathbb{R}^k} \frac{1}{2Mn} \sum_{m=1}^M \sum_{j=1}^n (\langle \boldsymbol{\theta}_1 - \boldsymbol{\theta}_*, \mathbf{x}_{m,j} \rangle + \langle \boldsymbol{\theta}_2, \mathbf{z}_m \rangle - b_m - \zeta_{m,j})^2.$$

Then, each client can achieve zero excess risk if there exists $\boldsymbol{\theta}_2$ such that $\langle \boldsymbol{\theta}_2, \mathbf{z}_m \rangle = b_m$ for all $m$, i.e. the side information is sufficiently expressive. Granted, learning such a $\boldsymbol{\theta}_2$ entails learning additional parameters if $k \geq 2$. But this is mitigated in settings with many clients since $\boldsymbol{\theta}_2$ is shared globally. *As a result, this example shows how learning a model of the form* (3) *can effectively leverage side information to achieve personalization.*

## C  Inductive Matrix Completion with FedAvg

### C.1  Further Background

Recall the server's population objective is:

$$\min_{\mathbf{U}, \mathbf{V}} \mathcal{L}(\mathbf{U}, \mathbf{V}) := \frac{1}{2M} \|\mathbf{U}\mathbf{V}^\top \mathbf{Z}^\top - \mathbf{M}_* \mathbf{Z}^\top\|_F^2 \tag{9}$$

where $\mathbf{M}_* = \mathbf{L}_* \mathbf{Z}^\top \in \mathbb{R}^{d \times M}$ is the ground-truth client-item ranking matrix, and $\bar{\mathbf{M}}_* \in \mathbb{R}^{d \times k}$. Note that the global loss $f$ can be written as the average of local losses $\mathcal{L}_m$:

$$\mathcal{L}(\mathbf{U}, \mathbf{V}) = \frac{1}{M} \sum_{m=1}^M \left\{ \mathcal{L}_m(\mathbf{U}, \mathbf{V}) := \frac{1}{2} \sum_{i=1}^d (\mathbf{e}_i^\top (\mathbf{U}\mathbf{V}^\top - \mathbf{M}_*) \mathbf{z}_m)^2 \right\}. \tag{10}$$

**Algorithm**. As mentioned in Section 3, the PerFedSI algorithm in this case is FedAvg on the IMC objective (10), with local updates being stochastic gradient descent steps on the local losses $\{\mathcal{L}_m\}_m$. In particular, the local updates by client $m$ on the $t$-th communication round are:

$$\mathbf{U}_{t,m,s+1} = \mathbf{U}_{t,m,s} - \eta \mathbf{e}_{t,m,s} \mathbf{e}_{t,m,s}^\top (\mathbf{U}_{t,m,s} \mathbf{V}_{t,m,s}^\top - \mathbf{M}_*) \mathbf{z}_m \mathbf{z}_m^\top \mathbf{V}_{t,m,s}, \tag{11}$$

$$\mathbf{V}_{t,m,s+1} = \mathbf{V}_{t,m,s} - \eta \mathbf{z}_m \mathbf{z}_m^\top (\mathbf{V}_{t,m,s} \mathbf{U}_{t,m,s}^\top - \mathbf{M}_*) \mathbf{e}_{t,m,s} \mathbf{e}_{t,m,s}^\top \mathbf{U}_{t,m,s}, \tag{12}$$

for $s + 1 \in [\tau]$, where each $\mathbf{e}_{t,m,s}$ is an independent sample from $\mathrm{Unif}(\{\mathbf{e}_1, \ldots, \mathbf{e}_d\})$, i.e. the uniform distribution over the standard basis vectors on $\mathbb{R}^d$. That is, each local update involves a *fresh sample*. Client $m$'s training dataset can be taken to be the set of samples observed throughout training, namely $D_m := \{(\mathbf{e}_{t,m,s}, \mathbf{e}_{t,m,s}^\top \mathbf{M}_* \mathbf{z}_m)\}_{t \in [T]_0, s \in [\tau-1]_0}$. We will show that the total number of rounds required to reach $\epsilon$-error is sufficiently small such that $|D_m| \ll d$ as long as $\epsilon \gg e^{-d}$.

Table 1: Summary of notations used in the analysis.

| Name | Description |
| --- | --- |
| $M$ | Number of clients |
| $m$ | Index over clients |
| $T$ | Number of communication rounds |
| $t$ | Index over communication rounds |
| $\tau$ | Number of local updates |
| $s$ | Index over local updates |
| $d$ | Number of items |
| $i$ | Index over items |
| $r$ | Rank of ground-truth matrix |
| $k$ | Dimension of side information |
| $\eta$ | Step size |
| $\mathbf{U}_{t,m,s}$ | Locally-updated $\mathbf{U}$ matrix after $t$ comm. rounds and $s$ local updates by client $m$ |
| $\hat{\mathbf{U}}_{t,m,s}\mathbf{R}_{t,m,s}$ | QR decomposition of $\mathbf{U}_{t,m,s}$ |
| $\mathbf{V}_{t,m,s}$ | Locally-updated $\mathbf{V}$ matrix after $t$ comm. rounds and $s$ local updates by client $m$ |
| $\hat{\mathbf{V}}_{t,m,s}\mathbf{R}'_{t,m,s}$ | QR decomposition of $\mathbf{V}_{t,m,s}$ |
| $\mathbf{M}_*\mathbf{Z}^\top$ | Ground-truth matrix $\in \mathbb{R}^{d\times M}$ |
| $\hat{\mathbf{X}}_*\mathbf{\Sigma}_*\hat{\mathbf{Y}}_*^\top$ | SVD of $\mathbf{M}_*\mathbf{Z}^\top$ |
| $\mathbf{Z}$ (resp. $\mathbf{z}_m$) | Side information matrix (resp. $m$-th row of $\mathbf{Z}$) |
| $\mathcal{L}(\mathbf{U},\mathbf{V})$ | Global population loss function (see (10)) |
| $\mathcal{L}_m(\mathbf{U},\mathbf{V})$ | Local population loss function (see (10)) |
| $\mathbf{e}_i$ | The $i$-th standard basis vector in $\mathbb{R}^d$ (deterministic) |
| $\mathbf{e}_{t,m,s}$ | i.i.d. sample from $\mathrm{Unif}(\{\mathbf{e}_1,\dots,\mathbf{e}_d\})$ by client $m$ on $s$-th local update on round $t$ |
| $\mathbf{E}_{t,m,s}$ | Local error $\mathbf{U}_{t,m,s}, \mathbf{V}_{t,m,s}^\top - \mathbf{U}_*\mathbf{V}_*$ |
| $\sigma_{1,*}$ | Maximum singular value of $\mathbf{U}_*\mathbf{V}_*^\top$ |
| $\sigma_{r,*}$ | Minimum singular value of $\mathbf{U}_*\mathbf{V}_*^\top$ |
| $\mu_z$ | Incoherence parameter for $\mathbf{Z}$ (see Assumption **??**) |
| $\mu_U$ | Incoherence parameter for $\mathbf{U}$ (see Assumption (**??**)) |
| $\mu_E$ | Incoherence parameter for $\mathbf{E} := \mathbf{U}\mathbf{V}^\top - \mathbf{M}_*$ (see Assumption (**??**)) |
| $\mu$ | $\max(\mu_z, \mu_U, \mu_E)$ |
| $\mathcal{S}_t, \mathcal{S}_{t,s}$ | $\sigma$-algebra induced by stochastic gradients up to round $t$, $s$-th local update on round $t$, respectively |
| $\mathbb{E}_t[\cdot]$ | $\mathbb{E}[\cdot\|\mathcal{S}_t]$ |
| $\mathcal{G}_{t,s}$ | Event that all global and local updates stay in good local regions (see (**??**)) up to the start of the $s$-th local updates on round $t$. |

Note that $\mathbf{U}_{t,m,0} := \mathbf{U}_t$ and $\mathbf{V}_{t,m,0} := \mathbf{V}_t$. For analysis purposes only, we define these local updates for all $m \in [M]$ on each round, even though only a subset of the clients participate in each round. Next, we make the following assumption on the manner in which the clients are sampled by the server.

**Assumption C.1.** Each selected client on each round is drawn independently from $\mathrm{Unif}(\{1,\dots,M\})$.

Note that Assumption C.1 allows for the same client to be selected twice on the same round. In this case, our analysis treats the local updates resulting from each selection as independent samples from the same random process. However, since $C$ is a constant and $M$ is often very large in practice, selecting the same client more than once on the same round is a low-probability event.

The global updates are then

$$\mathbf{U}_{t+1} = \frac{1}{C}\sum_{j\in\mathcal{B}_t}\mathbf{U}_{t,j,\tau},$$

$$\mathbf{V}_{t+1} = \frac{1}{C}\sum_{j\in\mathcal{B}_t}\mathbf{V}_{t,j,\tau}.$$

Note that each $j \in \mathcal{B}_t$ is a random variable, and by Assumption C.1, $j \sim \text{Unif}(\{1, \ldots, M\})$. By the linearity of expectation,

$$\mathbb{E}_t[\mathbf{U}_{t+1}] = \frac{1}{C} \sum_{j \in \mathcal{B}_t} \mathbb{E}_t[\mathbf{U}_{t,j,\tau}] = \frac{1}{C} \sum_{j \in \mathcal{B}_t} \frac{1}{M} \sum_{m=1}^{M} \mathbb{E}_t[\mathbf{U}_{t,m,\tau}] = \frac{1}{M} \sum_{m=1}^{M} \mathbb{E}_t[\mathbf{U}_{t,m,\tau}]. \quad (13)$$

**Remark C.2** (Privacy). To enhance the privacy of side information, one may add noise to each client's $\mathbf{V}_{t,m,\tau}$ before it is received by the server. We do not consider the effect of this noise on the algorithm's convergence in this analysis, and leave it for future work.

Recall Assumption 3.2 from the main paper. The scaling of $\mathbf{Z}$ outlined in Assumption 3.2 corresponds roughly to the case that each client embedding $\mathbf{z}_m$ is sampled from a multivariate Gaussian distribution with zero mean and identity covariance.

Now, we define *incoherence*, which is a key property that defines the events $\{\mathcal{A}_{t,s}\}_{t,s}$ (and hence, $\{\mathcal{G}_{t,s}\}_{t,s}$) and is used critically in matrix sensing analysis [14]. The importance of incoherence stems from the fact that the ground-truth matrix can only be recovered with an efficient sample size if it is non-aligned with the sampling vectors.

**Definition C.3** (Incoherence). A matrix $\mathbf{A} \in \mathbb{R}^{d \times r}$ is said to be $\mu$-incoherent if

$$\max_{i \in [d]} \|\mathbf{e}_i^\top \mathbf{A}\|_2 \leq \sqrt{\frac{\mu r}{d}} \|\mathbf{A}\|_2. \quad (14)$$

The event $\mathcal{A}_{t,s}$ entails that $\mathbf{U}_{t,m,s}$ and $\mathbf{E}_{t,m,s}$ are incoherent with respect to the standard basis. Likewise, Assumption 3.2 entails that $\mathbf{Z}$ is incoherent with respect to the standard basis. For ease of notation we denote $\mu := \max(\mu_U, \mu_E, \mu_z)$.

## C.2 Proof Sketch

The proof leverages that if $\mathcal{G}_{T,0}$ is satisfied, then the global updates remain in a favorable global region within which the objective is $\beta$-smooth and the Polyak-Lojasiewicz (PL) Inequality is satisfied with parameter $\gamma$, i.e. $\|\nabla \mathcal{L}(\mathbf{U}_t, \mathbf{V}_t)\|_F^2 \geq \gamma \mathcal{L}(\mathbf{U}_t, \mathbf{V}_t)$. Further, if $\mathcal{G}_{T,0}$ holds, then the client drift on each round is small, specifically $\left\| \sum_{s=0}^{\tau-1} \frac{d}{M} \sum_{m=1}^{M} \mathbb{E}_t[\nabla \mathcal{L}(\mathbf{U}_{t,m,s}; \mathbf{V}_{t,m,s})] - \tau \nabla \mathcal{L}(\mathbf{U}_t, \mathbf{V}_t) \right\|_F 1_{\mathcal{G}_{T,0}} = O(\eta \tau^2) \sqrt{\mathcal{L}(\mathbf{U}_t, \mathbf{V}_t)}$. Using these properties, we can show, conditioned on $\mathcal{G}_{T,0}$ the history up to time $t$,

$$\mathbb{E}_t[\mathcal{L}(\mathbf{U}_{t+1}, \mathbf{V}_{t+1})]$$
$$\leq \mathcal{L}(\mathbf{U}_t, \mathbf{V}_t) + \langle \nabla \mathcal{L}(\mathbf{U}_t, \mathbf{V}_t), \mathbb{E}_t[\mathbf{U}_{t+1}; \mathbf{V}_{t+1}] - [\mathbf{U}_t; \mathbf{V}_t] \rangle$$
$$+ \frac{\beta}{2} \mathbb{E}_t[\|\mathbf{U}_{t+1} - \mathbf{U}_t\|^2 + \|\mathbf{V}_{t+1} - \mathbf{V}_t\|^2]$$
$$\leq \mathcal{L}(\mathbf{U}_t, \mathbf{V}_t) - \frac{\eta \tau}{d} \|\nabla \mathcal{L}(\mathbf{U}_t, \mathbf{V}_t)\|_F^2$$
$$+ \frac{\eta}{d} \|\nabla \mathcal{L}(\mathbf{U}_t, \mathbf{V}_t)\|_F \left\| \sum_{s=0}^{\tau-1} \frac{d}{M} \sum_{m=1}^{M} \mathbb{E}_t[\nabla \mathcal{L}(\mathbf{U}_{t,m,s}; \mathbf{V}_{t,m,s})] - \tau \nabla \mathcal{L}(\mathbf{U}_t, \mathbf{V}_t) \right\|_F$$
$$+ O(\frac{\eta^2 \tau^2}{d}) \mathcal{L}(\mathbf{U}_t, \mathbf{V}_t)$$
$$\leq (1 - \frac{\eta \tau}{d} \gamma + O(\frac{\eta^2 \tau^2}{d})) \mathcal{L}(\mathbf{U}_t, \mathbf{V}_t),$$

which completes the proof by choosing a sufficiently small $\alpha$. We have omitted the dependence on $k$ here for brevity. The full proof is given in the following subsection.

## C.3 Formal Theorem Statement and Proof

**Theorem C.4** (Formal). *Suppose that $\eta \leq \frac{\sigma_{r,*}}{132 k^{3/2} r \tau \mu^2 \sigma_{1,*}^2 c^5 c_z^4}$ and Assumptions 3.1, 3.2, and C.1 are satisfied. Then PerFedSI run on the matrix completion with side information problem with a constant $C \geq 1$ clients participating per round and $\tau$ local updates per round, converges linearly in expectation to the ground-truth matrix, in particular*

$$\mathbb{E}[\mathcal{L}(\mathbf{U}_t, \mathbf{V}_t) 1_{\mathcal{G}_{T-1,\tau}}] \leq (1 - 0.5 \eta \tau \frac{\sigma_{r,*}}{c c_z^3 dk})^{T-1} \mathcal{L}(\mathbf{U}_0, \mathbf{V}_0). \quad (15)$$

516  Throughout the proof, let $\hat{\mathbf{U}}$ denote the left singular vectors of the matrix $\mathbf{U}$, for any matrix $\mathbf{U}$, and
517  let $\mathcal{P}_{\hat{\mathbf{U}}}$ denote the orthogonal projection onto $\mathrm{col}(\hat{\mathbf{U}})$, and $\mathcal{P}_{\hat{\mathbf{U}}_{\perp}}$ denote the orthogonal projection
518  onto the subspace perpendicular to $\mathrm{col}(\hat{\mathbf{U}})$.

519  The next two lemmas are adaptations of Lemmas C.2, C.3, and C.4 in [15] to our setting with side
520  information.

521  **Lemma C.5** (PL Condition)**.** *Within the region* $\{(\mathbf{U}, \mathbf{V}) : \mathbf{U} \in \mathbb{R}^{d \times r}, \mathbf{V} \in \mathbb{R}^{k \times r}, \sigma_{\min}(\mathbf{U}_*^\top \mathbf{U}) \geq$
522  $\gamma, \sigma_{\min}(\mathbf{V}_*^\top \mathbf{V}) \geq \gamma\}$, *the function* $\mathcal{L}(\mathbf{U}, \mathbf{V}) := \frac{1}{2M}\|(\mathbf{U}\mathbf{V}^\top - \mathbf{U}_*\mathbf{V}_*^\top)\mathbf{Z}^\top\|_F^2$ *satisfies*

$$\|\nabla \mathcal{L}(\mathbf{U}, \mathbf{V})\|_F^2 \geq \gamma \mathcal{L}(\mathbf{U}, \mathbf{V}), \tag{16}$$

523  *where* $\gamma := \frac{\sigma_{r,*}}{cc_z^3}$.

524  *Proof.* For the gradient with respect to $\mathbf{U}$, we have

$$
\begin{aligned}
\|\nabla_{\mathbf{U}} \mathcal{L}(\mathbf{U}, \mathbf{V})\|_F^2 &= \frac{1}{M^2}\|(\mathbf{U}\mathbf{V}^\top - \mathbf{M}_*)\mathbf{Z}^\top \mathbf{Z}\mathbf{V}\|_F^2 \\
&= \frac{1}{M^2}\|\mathcal{P}_{\hat{\mathbf{U}}}((\mathbf{U}\mathbf{V}^\top - \mathbf{M}_*)\mathbf{Z}^\top \mathbf{Z}\mathbf{V})\|_F^2 + \frac{1}{M^2}\|\mathcal{P}_{\hat{\mathbf{U}}_{\perp}}(\mathbf{U}\mathbf{V}^\top - \mathbf{M}_*)\mathbf{Z}^\top \mathbf{Z}\mathbf{V}\|_F^2 \\
&= \frac{1}{M^2}\|\mathcal{P}_{\hat{\mathbf{U}}}(\mathbf{U}\mathbf{V}^\top - \mathbf{M}_*)\mathbf{Z}^\top \mathbf{Z}\mathbf{V}\|_F^2 + \frac{1}{M^2}\|\mathcal{P}_{\hat{\mathbf{U}}_{\perp}}\mathbf{M}_*\mathbf{Z}^\top \mathbf{Z}\mathbf{V}\|_F^2 \\
&\geq \frac{\sigma_{\min}^2(\mathbf{Z}\mathbf{V})}{M^2}\|\mathcal{P}_{\hat{\mathbf{U}}}(\mathbf{U}\mathbf{V}^\top - \mathbf{M}_*)\mathbf{Z}^\top \mathcal{P}_{\mathbf{Z}\mathbf{V}}\|_F^2 + \frac{\sigma_{\min}^2(\hat{\mathbf{Y}}_*^\top \mathbf{Z}\mathbf{V})}{M^2}\|\mathcal{P}_{\hat{\mathbf{U}}_{\perp}}\hat{\mathbf{X}}_*\mathbf{\Sigma}_*\|_F^2 \\
&\geq \frac{\sigma_{\min}^2(\mathbf{Z}\mathbf{V})}{M^2}\|\mathcal{P}_{\hat{\mathbf{U}}}(\mathbf{U}\mathbf{V}^\top - \mathbf{M}_*)\mathbf{Z}^\top \mathcal{P}_{\mathbf{Z}\mathbf{V}}\|_F^2 + \frac{\sigma_{\min}^2((\hat{\mathbf{Y}}_*^\top \mathbf{Z}\mathbf{V})}{M^2}\|\mathcal{P}_{\hat{\mathbf{U}}_{\perp}}\mathbf{M}_*\mathbf{Z}^\top\|_F^2
\end{aligned}
\tag{17}
$$

525  For the gradient with respect to $\mathbf{V}$, we have

$$
\begin{aligned}
\|\nabla_{\mathbf{V}} \mathcal{L}(\mathbf{U}, \mathbf{V})\|_F^2 &= \frac{1}{M^2}\|\mathbf{U}^\top(\mathbf{U}\mathbf{V}^\top - \mathbf{M}_*)\mathbf{Z}^\top \mathbf{Z}\|_F^2 \\
&\geq \frac{\sigma_{\min}^4(\mathbf{Z})}{M^2}\|\mathbf{U}^\top(\mathbf{U}\mathbf{V}^\top - \mathbf{M}_*)\|_F^2 \\
&\geq \frac{\sigma_{\min}^4(\mathbf{Z})}{M^2\|\mathbf{Z}\|_2^2}\|\mathbf{U}^\top(\mathbf{U}\mathbf{V}^\top - \mathbf{M}_*)\mathbf{Z}^\top\|_F^2.
\end{aligned}
\tag{18}
$$

526  where (18) follows since for any matrix $\mathbf{A}$ with commensurate dimension,

$$\|\mathbf{A}\mathbf{Z}^\top\|_F^2 \leq \|\mathbf{A}\|_F^2\|\mathbf{Z}\|_2^2.$$

527  Next, we have

$$
\begin{aligned}
&\|\mathbf{U}^\top(\mathbf{U}\mathbf{V}^\top - \mathbf{M}_*)\mathbf{Z}^\top\|_F^2 \\
&= \|\mathbf{U}^\top(\mathbf{U}\mathbf{V}^\top - \mathbf{M}_*)\mathbf{Z}^\top \mathcal{P}_{\mathbf{Z}\mathbf{V}}\|_F^2 + \|\mathbf{U}^\top(\mathbf{U}\mathbf{V}^\top - \mathbf{M}_*)\mathbf{Z}^\top \mathcal{P}_{(\mathbf{Z}\mathbf{V})_{\perp}}\|_F^2 \\
&\geq \sigma_{\min}^2(\mathbf{U})\|\mathcal{P}_{\hat{\mathbf{U}}}(\mathbf{U}\mathbf{V}^\top - \mathbf{M}_*)\mathbf{Z}^\top \mathcal{P}_{\mathbf{Z}\mathbf{V}}\|_F^2 + \|\mathbf{U}^\top \mathbf{M}_*\mathbf{Z}^\top \mathcal{P}_{(\mathbf{Z}\mathbf{V})_{\perp}}\|_F^2 \\
&\geq \sigma_{\min}^2(\mathbf{U})\|\mathcal{P}_{\hat{\mathbf{U}}}(\mathbf{U}\mathbf{V}^\top - \mathbf{M}_*)\mathbf{Z}^\top \mathcal{P}_{\mathbf{Z}\mathbf{V}}\|_F^2 + \sigma_{\min}^2(\mathbf{U}^\top \hat{\mathbf{U}}_*)\|\mathbf{\Sigma}_*\mathbf{Y}_*^\top \mathcal{P}_{(\mathbf{Z}\mathbf{V})_{\perp}}\|_F^2 \\
&\geq \sigma_{\min}^2(\mathbf{U})\|\mathcal{P}_{\hat{\mathbf{U}}}(\mathbf{U}\mathbf{V}^\top - \mathbf{M}_*)\mathbf{Z}^\top \mathcal{P}_{\mathbf{Z}\mathbf{V}}\|_F^2 + \sigma_{\min}^2(\mathbf{U}^\top \hat{\mathbf{U}}_*)\|\mathbf{M}_*\mathbf{Z}^\top \mathcal{P}_{(\mathbf{Z}\mathbf{V})_{\perp}}\|_F^2
\end{aligned}
\tag{19}
$$

528  Combining (17), (18) and (19) yields

$$
\begin{aligned}
\|\nabla \mathcal{L}(\mathbf{U}, \mathbf{V})\|_F^2 &= \|\nabla_{\mathbf{U}} \mathcal{L}(\mathbf{U}, \mathbf{V})\|_F^2 + \|\nabla_{\mathbf{V}} \mathcal{L}(\mathbf{U}, \mathbf{V})\|_F^2 \\
&\geq \frac{\sigma_{\min}^2(\mathbf{Z}\mathbf{V})}{M^2}\|\mathcal{P}_{\hat{\mathbf{U}}}(\mathbf{U}\mathbf{V}^\top - \mathbf{M}_*)\mathbf{Z}^\top \mathcal{P}_{\mathbf{Z}\mathbf{V}}\|_F^2 + \frac{\sigma_{\min}^2(\hat{\mathbf{Y}}_*^\top \mathbf{Z}\mathbf{V})}{M^2}\|\mathcal{P}_{\hat{\mathbf{U}}_{\perp}}\mathbf{M}_*\mathbf{Z}^\top\|_F^2 \\
&\quad + \frac{\sigma_{\min}^4(\mathbf{Z})}{M^2\|\mathbf{Z}\|_2^2}\sigma_{\min}^2(\mathbf{U})\|\mathcal{P}_{\hat{\mathbf{U}}}(\mathbf{U}\mathbf{V}^\top - \mathbf{M}_*)\mathbf{Z}^\top \mathcal{P}_{\mathbf{Z}\mathbf{V}}\|_F^2 \\
&\quad + \frac{\sigma_{\min}^4(\mathbf{Z})}{M^2\|\mathbf{Z}\|_2^2}\sigma_{\min}^2(\mathbf{U}^\top \hat{\mathbf{U}}_*)\|\mathbf{M}_*\mathbf{Z}^\top \mathcal{P}_{(\mathbf{Z}\mathbf{V})_{\perp}}\|_F^2 \\
&\geq \frac{1}{M}\min\left(\sigma_{\min}^2(\hat{\mathbf{Y}}_*^\top \mathbf{Z}\mathbf{V}), \frac{1}{c_z^2}\sigma_{\min}^2(\mathbf{Z})\sigma_{\min}^2(\hat{\mathbf{U}}_*^\top \mathbf{U})\right)\mathcal{L}(\mathbf{U}, \mathbf{V}) \tag{20} \\
&\geq \frac{\sigma_{r,*}}{cc_z^3}\mathcal{L}(\mathbf{U}, \mathbf{V}) \tag{21}
\end{aligned}
$$

529    using that $c_z \geq 1$ and $\sigma_{\min}(\hat{\mathbf{Y}}_*^\top \mathbf{Z}\mathbf{V}) \geq \sqrt{\sigma_{r,*}/c}$.        $\square$

530    **Lemma C.6** (Smoothness). *Within the region* $\mathcal{D}_\beta := \{(\mathbf{U},\mathbf{V}) : \mathbf{U} \in \mathbb{R}^{d \times r}, \mathbf{V} \in \mathbb{R}^{k \times r}, \|\mathbf{U}\|_2 \leq$
531    $c\sqrt{\sigma_{1,*}}, \|\mathbf{V}\|_2 \leq c\sqrt{\sigma_{1,*}}\}$, *the function* $\mathcal{L}(\mathbf{U},\mathbf{V}) := \frac{1}{2M}\|(\mathbf{U}\mathbf{V}^\top - \mathbf{U}_*\mathbf{V}_*^\top)\mathbf{Z}^\top\|_F^2$ *satisfies*

$$\|\nabla\mathcal{L}(\mathbf{U}_1,\mathbf{V}_1) - \nabla\mathcal{L}(\mathbf{U}_2,\mathbf{V}_2)\|_F^2 \leq \frac{\beta^2}{2}(\|\mathbf{U}_1 - \mathbf{U}_2\|_F^2 + \|\mathbf{V}_1 - \mathbf{V}_2\|_F^2). \tag{22}$$

532    *where* $\beta^2 := c_z^2\left(32c^4 + 4\right)\sigma_{1,*}^2$.

533    *Proof.* Note that

$$\|\nabla\mathcal{L}(\mathbf{U}_1,\mathbf{V}_1) - \nabla\mathcal{L}(\mathbf{U}_2,\mathbf{V}_2)\|_F^2$$
$$= \|\nabla_{\mathbf{U}}\mathcal{L}(\mathbf{U}_1,\mathbf{V}_1) - \nabla_{\mathbf{U}}\mathcal{L}(\mathbf{U}_2,\mathbf{V}_2)\|_F^2 + \|\nabla_{\mathbf{V}}\mathcal{L}(\mathbf{U}_1,\mathbf{V}_1) - \nabla_{\mathbf{V}}\mathcal{L}(\mathbf{U}_2,\mathbf{V}_2)\|_F^2$$
$$= \frac{1}{M^2}\|(\mathbf{U}_1\mathbf{V}_1^\top - \mathbf{U}_*\mathbf{V}_*^\top)\mathbf{Z}^\top\mathbf{Z}\mathbf{V}_1 - (\mathbf{U}_2\mathbf{V}_2^\top - \mathbf{U}_*\mathbf{V}_*^\top)\mathbf{Z}^\top\mathbf{Z}\mathbf{V}_2\|_F^2$$
$$+ \frac{1}{M^2}\|\mathbf{U}_1^\top(\mathbf{U}_1\mathbf{V}_1^\top - \mathbf{U}_*\mathbf{V}_*^\top)\mathbf{Z}^\top\mathbf{Z} - \mathbf{U}_2^\top(\mathbf{U}_2\mathbf{V}_2^\top - \mathbf{U}_*\mathbf{V}_*^\top)\mathbf{Z}^\top\mathbf{Z}\|_F^2 \tag{23}$$

534    For the first term, by repeatedly applying the inequalities $\|\mathbf{A} + \mathbf{B}\|_F^2 \leq 2(\|\mathbf{A}\|_F^2 + \|\mathbf{B}\|_F^2)$ and
535    $\|\mathbf{A}\mathbf{B}\|_F \leq \|\mathbf{A}\|_2\|\mathbf{B}\|_F$ we have

$$\|(\mathbf{U}_1\mathbf{V}_1^\top - \mathbf{M}_*)\mathbf{Z}^\top\mathbf{Z}\mathbf{V}_1 - (\mathbf{U}_2\mathbf{V}_2^\top - \mathbf{M}_*)\mathbf{Z}^\top\mathbf{Z}\mathbf{V}_2\|_F^2$$
$$\leq 2\|\mathbf{U}_1\mathbf{V}_1^\top\mathbf{Z}^\top\mathbf{Z}\mathbf{V}_1 - \mathbf{U}_2\mathbf{V}_2^\top\mathbf{Z}^\top\mathbf{Z}\mathbf{V}_2\|_F^2 + 2\|\mathbf{M}_*\mathbf{Z}^\top\mathbf{Z}\mathbf{V}_1 - \mathbf{M}_*\mathbf{Z}^\top\mathbf{Z}\mathbf{V}_2\|_F^2$$
$$\leq 4\|\mathbf{U}_1\mathbf{V}_1^\top\mathbf{Z}^\top\mathbf{Z}(\mathbf{V}_1 - \mathbf{V}_2)\|_F^2 + 4\|(\mathbf{U}_1\mathbf{V}_1^\top - \mathbf{U}_2\mathbf{V}_2^\top)\mathbf{Z}^\top\mathbf{Z}\mathbf{V}_2\|_F^2 + 2\|\mathbf{M}_*\|_2^2\|\mathbf{Z}\|_2^4\|\mathbf{V}_1 - \mathbf{V}_2\|_F^2$$
$$\leq 4\|\mathbf{U}_1\mathbf{V}_1^\top\|_2^2\|\mathbf{Z}\|_2^4\|\mathbf{V}_1 - \mathbf{V}_2\|_F^2 + 8(\|\mathbf{U}_1(\mathbf{V}_1^\top - \mathbf{V}_2^\top)\|_F^2 + \|(\mathbf{U}_1 - \mathbf{U}_2)\mathbf{V}_2^\top\|_F^2)\|\mathbf{Z}\|_2^4\|\mathbf{V}_2\|_2^2$$
$$+ 2\|\mathbf{M}_*\|_2^2\|\mathbf{Z}\|_2^4\|\mathbf{V}_1 - \mathbf{V}_2\|_F^2$$
$$\leq \|\mathbf{Z}\|_2^2\left(12c^4\sigma_{1,*}^2 + 2\sigma_{1,*}^2\right)\|\mathbf{V}_1 - \mathbf{V}_2\|_F^2 + 4c^4\sigma_{1,*}^2\|\mathbf{Z}\|_2^2\|\mathbf{U}_1 - \mathbf{U}_2\|_F^2 \tag{24}$$

536    A similar argument for the second term in (23) yields

$$\|\nabla\mathcal{L}(\mathbf{U}_1,\mathbf{V}_1) - \nabla\mathcal{L}(\mathbf{U}_2,\mathbf{V}_2)\|_F^2 \leq \left(16c^4 + 2\right)\frac{\sigma_{1,*}^2}{M^2}\|\mathbf{Z}\|_2^4\left(\|\mathbf{V}_1 - \mathbf{V}_2\|_F^2 + \|\mathbf{U}_1 - \mathbf{U}_2\|_F^2\right)$$
$$\leq c_z^2\left(16c^4 + 2\right)\sigma_{1,*}^2\left(\|\mathbf{V}_1 - \mathbf{V}_2\|_F^2 + \|\mathbf{U}_1 - \mathbf{U}_2\|_F^2\right),$$

537    where the last inequality follows by Assumption 3.2.        $\square$

538    **Lemma C.7** (Bound on second-order error). *For any* $t$,

$$\mathbb{E}_t[\|\mathbf{U}_{t+1} - \mathbf{U}_t\|_F^2 1_{\mathcal{G}_{t,\tau}}] \leq \eta^2\tau^2 k\left(\frac{1}{Cd} + \frac{1}{d^2}\right)c\mu_z\sigma_{1,*}\mathcal{L}(\mathbf{U}_t,\mathbf{V}_t)1_{\mathcal{G}_{t,0}}$$

$$\mathbb{E}_t[\|\mathbf{V}_{t+1} - \mathbf{V}_t\|_F^2 1_{\mathcal{G}_{t,\tau}}] \leq \eta^2\tau^2 kr\frac{(c + \mu_U)\mu_z\sigma_{1,*}}{d^2}\mathcal{L}(\mathbf{U}_t,\mathbf{V}_t)1_{\mathcal{G}_{t,0}}$$

539    *Proof.* We have

$$\mathbb{E}_t[\|\mathbf{U}_{t+1} - \mathbf{U}_t\|_F^2 1_{\mathcal{G}_{t,\tau}}]$$
$$= \eta^2\mathbb{E}_t\left[\left\|\frac{1}{C}\sum_{s=0}^{\tau-1}\sum_{j\in\mathcal{B}_t}\nabla_{\mathbf{U}}\hat{\mathcal{L}}_m(\mathbf{U}_{t,m,s},\mathbf{V}_{t,m,s},\mathbf{e}_{t,m,s})1_{\mathcal{G}_{t,\tau}}\right\|_F^2\right]$$
$$= \eta^2\mathbb{E}_t\left[\left\|\frac{1}{C}\sum_{s=0}^{\tau-1}\sum_{j\in\mathcal{B}_t}\mathbf{e}_{t,m,s}\mathbf{e}_{t,m,s}^\top(\mathbf{U}_{t,m,s}\mathbf{V}_{t,m,s}^\top - \mathbf{U}_*\mathbf{V}_*^\top)\mathbf{z}_j\mathbf{z}_j^\top\mathbf{V}_{t,m,s}1_{\mathcal{G}_{t,\tau}}\right\|_F^2\right]$$
$$\leq \eta^2\tau^2\max_{s\in\{0,\ldots,\tau-1\}}\mathbb{E}_t\left[\left\|\frac{1}{C}\sum_{j\in\mathcal{B}_t}\mathbf{e}_{t,m,s}\mathbf{e}_{t,m,s}^\top(\mathbf{U}_{t,m,s}\mathbf{V}_{t,m,s}^\top - \mathbf{U}_*\mathbf{V}_*^\top)\mathbf{z}_j\mathbf{z}_j^\top\mathbf{V}_{t,m,s}1_{\mathcal{G}_{t,\tau}}\right\|_F^2\right]$$
$$\tag{25}$$

540     Let $\mathbf{A}_{t,m,s} := (\mathbf{U}_{t,m,s}\mathbf{V}_{t,m,s}^\top - \mathbf{U}_*\mathbf{V}_*^\top)\mathbf{z}_j\mathbf{z}_j^\top\mathbf{V}_{t,m,s}$. We obtain for any $s \in \{0,\dots,\tau-1\}$,

$$\mathbb{E}_t\left[\left\|\frac{1}{C}\sum_{j\in\mathcal{B}_t}\mathbf{e}_{t,m,s}\mathbf{e}_{t,m,s}^\top(\mathbf{U}_{t,m,s}\mathbf{V}_{t,m,s}^\top - \mathbf{U}_*\mathbf{V}_*^\top)\mathbf{z}_j\mathbf{z}_j^\top\mathbf{V}_{t,m,s}1_{\mathcal{G}_{t,\tau}}\right\|_F^2\right]$$

$$\leq \mathbb{E}_t\left[\left\|\frac{1}{C}\sum_{j\in\mathcal{B}_t}\mathbf{e}_{t,m,s}\mathbf{e}_{t,m,s}^\top\mathbf{A}_{t,m,s}1_{\mathcal{G}_{t,s}}\right\|_F^2\right]$$

$$= \mathbb{E}_t\left[\mathrm{Tr}\left(\left(\frac{1}{C}\sum_{j\in\mathcal{B}_t}\mathbf{e}_{t,m,s}\mathbf{e}_{t,m,s}^\top\mathbf{A}_{t,m,s}1_{\mathcal{G}_{t,s}}\right)^\top\left(\frac{1}{C}\sum_{j\in\mathcal{B}_t}\mathbf{e}_{t,m,s}\mathbf{e}_{t,m,s}^\top\mathbf{A}_{t,m,s}1_{\mathcal{G}_{t,s}}\right)\right)\right]$$

$$= \mathrm{Tr}\left(\frac{1}{C^2}\sum_{j\in\mathcal{B}_t}\sum_{j'\in\mathcal{B}_t\backslash j}\mathbb{E}_t\left[\mathbf{A}_{t,m,s}^\top\mathbf{e}_{t,m,s}\mathbf{e}_{t,m,s}^\top\mathbf{e}_{t,m',s}\mathbf{e}_{t,m',s}^\top\mathbf{A}_{t,m',s}1_{\mathcal{G}_{t,s}}\right]\right)$$

$$+ \mathrm{Tr}\left(\frac{1}{C^2}\sum_{j\in\mathcal{B}_t}\mathbb{E}_t\left[\mathbf{A}_{t,m,s}^\top\mathbf{e}_{t,m,s}\mathbf{e}_{t,m,s}^\top\mathbf{A}_{t,m,s}1_{\mathcal{G}_{t,s}}\right]\right) \tag{26}$$

$$= \mathrm{Tr}\left(\frac{1}{C^2}\sum_{j\in\mathcal{B}_t}\sum_{j'\in\mathcal{B}_t\backslash j}\mathbb{E}_t\left[\mathbf{A}_{t,m,s}^\top\mathbf{e}_{t,m,s}\mathbf{e}_{t,m,s}^\top 1_{\mathcal{G}_{t,s}}\right]\mathbb{E}_t\left[\mathbf{e}_{t,m',s}\mathbf{e}_{t,m',s}^\top\mathbf{A}_{t,m',s}1_{\mathcal{G}_{t,s}}\right]\right)$$

$$+ \mathrm{Tr}\left(\frac{1}{C^2}\sum_{j\in\mathcal{B}_t}\mathbb{E}_t\left[\mathbf{A}_{t,m,s}^\top\mathbf{e}_{t,m,s}\mathbf{e}_{t,m,s}^\top\mathbf{A}_{t,m,s}1_{\mathcal{G}_{t,s}}\right]\right) \tag{27}$$

541     where

$$\mathbb{E}_t\left[\mathbf{A}_{t,m,s}^\top\mathbf{e}_{t,m,s}\mathbf{e}_{t,m,s}^\top 1_{\mathcal{G}_{t,s}}\right] = \frac{1}{dM}\sum_{m=1}^M\mathbb{E}_t[\mathbf{A}_{t,m,s}^\top 1_{\mathcal{G}_{t,s}}] \tag{28}$$

542     since $\mathbf{e}_{t,m,s}$ is independent of $\mathbf{A}_{t,m,s}$ and $\mathbb{E}[\mathbf{e}_{t,m,s}\mathbf{e}_{t,m,s}^\top] = \frac{1}{d}\mathbf{I}_d$. Thus

$$\mathrm{Tr}\left(\frac{1}{C^2}\sum_{j\in\mathcal{B}_t}\sum_{j'\in\mathcal{B}_t\backslash j}\mathbb{E}_t\left[\mathbf{A}_{t,m,s}^\top\mathbf{e}_{t,m,s}\mathbf{e}_{t,m,s}^\top 1_{\mathcal{G}_{t,s}}\right]\mathbb{E}_t\left[\mathbf{e}_{t,m',s}\mathbf{e}_{t,m',s}^\top\mathbf{A}_{t,m',s}1_{\mathcal{G}_{t,s}}\right]\right)$$

$$= \frac{C-1}{Cd^2M^2}\mathrm{Tr}\left(\sum_{m=1}^M\sum_{m'=1}^M\mathbb{E}_t\left[\mathbf{A}_{t,m,s}^\top 1_{\mathcal{G}_{t,s}}\right]\mathbb{E}_t\left[\mathbf{A}_{t,m',s}1_{\mathcal{G}_{t,s}}\right]\right)$$

$$= \frac{C-1}{Cd^2M^2}\mathrm{Tr}\left(\sum_{m=1}^M\sum_{m'=1}^M\mathbb{E}_t\left[\mathbf{V}_{t,m,s}^\top\mathbf{z}_m\mathbf{z}_m^\top\mathbf{E}_{t,m,s}^\top 1_{\mathcal{G}_{t,s}}\right]\mathbb{E}_t\left[\mathbf{E}_{t,m',s}\mathbf{z}_{m'}\mathbf{z}_{m'}^\top\mathbf{V}_{t,m',s}1_{\mathcal{G}_{t,s}}\right]\right)$$

$$\leq \frac{\mu_z c\sigma_{1,*}k}{d^2}\mathcal{L}(\mathbf{U}_t,\mathbf{V}_t)1_{\mathcal{G}_{t,0}} \tag{29}$$

543     where (29) follows since if $\mathcal{G}_{t,s}$ holds, then $\max_{m\in[M]}\|\mathbf{z}_m\|_2 \leq \sqrt{\mu_z k}$, $\max_{m\in[M]}\|\mathbf{V}_{t,m,s}\|_2 \leq$
544     $\sqrt{c\sigma_{1,*}}$, and $\max_{m\in[M]}\|\mathbf{E}_{t,m,s}\mathbf{z}_m\|_2 = \sqrt{\mathcal{L}_m(\mathbf{U}_{t,m,s},\mathbf{V}_{t,m,s})} \leq \sqrt{\mathcal{L}(\mathbf{U}_t,\mathbf{V}_t)}$, and $1_{\mathcal{G}_{t,s}} \leq 1_{\mathcal{G}_{t,0}}$.
545     For the second term in (27),

$$\mathbb{E}_t\left[\mathbf{A}_{t,m,s}^\top\mathbf{e}_{t,m,s}\mathbf{e}_{t,m,s}^\top\mathbf{A}_{t,m,s}1_{\mathcal{G}_{t,s}}\right] = \frac{1}{M}\sum_{m=1}^M\mathbb{E}_t\left[\mathbf{A}_{t,m,s}^\top\mathbf{e}_{t,m,s}\mathbf{e}_{t,m,s}^\top\mathbf{A}_{t,m,s}1_{\mathcal{G}_{t,s}}\right]$$

$$= \frac{1}{dM}\sum_{m=1}^M\sum_{i=1}^d\mathbb{E}_t\left[\mathbf{A}_{t,m,s}^\top\mathbf{e}_i\mathbf{e}_i^\top\mathbf{A}_{t,m,s}1_{\mathcal{G}_{t,s}}\right]$$

$$= \frac{1}{dM}\sum_{m=1}^M\sum_{i=1}^d\mathbb{E}_t\left[(\mathbf{e}_i^\top\mathbf{E}_{t,m,s}\mathbf{z}_m)^2\mathbf{V}_{t,m,s}^\top\mathbf{z}_m\mathbf{z}_m^\top\mathbf{V}_{t,m,s}1_{\mathcal{G}_{t,s}}\right]$$

546 which means

$$
\begin{aligned}
&\mathrm{Tr}\left(\frac{1}{C^2}\sum_{j\in\mathcal{B}_t}\mathbb{E}_t\left[\mathbf{A}_{t,m,s}^\top\mathbf{e}_{t,m,s}\mathbf{e}_{t,m,s}^\top\mathbf{A}_{t,m,s}1_{\mathcal{G}_{t,s}}\right]\right)\\
&=\frac{1}{CdM}\sum_{m=1}^M\sum_{i=1}^d\mathbb{E}_t\left[(\mathbf{e}_i^\top\mathbf{E}_{t,m,s}\mathbf{z}_m)^2\,\mathrm{Tr}\left(\mathbf{V}_{t,m,s}^\top\mathbf{z}_m\mathbf{z}_m^\top\mathbf{V}_{t,m,s}1_{\mathcal{G}_{t,s}}\right)\right]\\
&\leq\frac{c\sigma_{1,*}\mu_z k}{Cd}\frac{1}{M}\sum_{m=1}^M\sum_{i=1}^d\mathbb{E}_t\left[(\mathbf{e}_i^\top\mathbf{E}_{t,m,s}\mathbf{z}_m)^2 1_{\mathcal{G}_{t,s}}\right]\\
&=\frac{c\sigma_{1,*}\mu_z k}{Cd}\mathbb{E}_t\left[\frac{1}{M}\sum_{m=1}^M\mathcal{L}_m(\mathbf{U}_{t,m,s},\mathbf{V}_{t,m,s})1_{\mathcal{G}_{t,s}}\right]\\
&\leq\frac{c\sigma_{1,*}\mu_z k}{Cd}\mathcal{L}(\mathbf{U}_t,\mathbf{V}_t)1_{\mathcal{G}_{t,0}}
\end{aligned}
\tag{30}
$$

547 Combining (27) with (29) and (30) yields

$$
\mathbb{E}_t[\|\mathbf{U}_{t+1}-\mathbf{U}_t\|_F^2 1_{\mathcal{G}_{t,\tau}}]\leq\eta^2\tau^2 k\left(\frac{1}{Cd}+\frac{1}{d^2}\right)c\mu_z\sigma_{1,*}\mathcal{L}(\mathbf{U}_t,\mathbf{V}_t)1_{\mathcal{G}_{t,0}}
\tag{31}
$$

548 The bound on $\mathbb{E}_t[\|\mathbf{V}_{t+1}-\mathbf{V}_t\|_F^2 1_{\mathcal{G}_{t,\tau}}]$ follows by a similar argument, but with some notable
549 differences. We obtain:

$$
\begin{aligned}
&\mathbb{E}_t[\|\mathbf{V}_{t+1}-\mathbf{V}_t\|_F^2 1_{\mathcal{G}_{t,\tau}}]\\
&\leq\max_{s\in\{0,\tau-1\}}\frac{\eta^2\tau^2}{C^2}\sum_{j,j'\in\mathcal{B}_t,j'\neq j}\mathrm{Tr}\left(\mathbb{E}_t\left[\mathbf{U}_{t,m,s}^\top\mathbf{e}_{t,m,s}\mathbf{e}_{t,m,s}^\top\mathbf{E}_{t,m,s}\mathbf{z}_m\mathbf{z}_m^\top 1_{\mathcal{G}_{t,s}}\right]\times\right.\\
&\qquad\qquad\qquad\qquad\qquad\left.\mathbb{E}_t\left[\mathbf{z}_{m'}\mathbf{z}_{m'}^\top\mathbf{E}_{t,m',s}^\top\mathbf{e}_{t,m',s}\mathbf{e}_{t,m',s}^\top\mathbf{U}_{t,m',s}1_{\mathcal{G}_{t,s}}\right]\right)\\
&\quad+\frac{\eta^2\tau^2}{C^2}\sum_{j\in\mathcal{B}_t}\mathrm{Tr}\left(\mathbb{E}_t\left[\|\mathbf{z}_m\|_2^2\mathbf{U}_{t,m,s}^\top\mathbf{e}_{t,m,s}\mathbf{e}_{t,m,s}^\top\mathbf{E}_{t,m,s}\mathbf{z}_m\mathbf{z}_m^\top\mathbf{E}_{t,m,s}^\top\mathbf{e}_{t,m,s}\mathbf{e}_{t,m,s}^\top\mathbf{U}_{t,m,s}1_{\mathcal{G}_{t,s}}\right]\right)
\end{aligned}
\tag{32}
$$

550 For the first term, we have

$$
\mathbb{E}_t\left[\mathbf{z}_m\mathbf{z}_m^\top\mathbf{E}_{t,m,s}^\top\mathbf{e}_{t,m,s}\mathbf{e}_{t,m,s}^\top\mathbf{U}_{t,m,s}1_{\mathcal{G}_{t,s}}\right]=\frac{1}{dM}\sum_{m=1}^M\mathbb{E}_t[\mathbf{z}_m\mathbf{z}_m^\top\mathbf{E}_{t,m,s}^\top\mathbf{U}_{t,m,s}1_{\mathcal{G}_{t,s}}]
\tag{33}
$$

551 thus

$$
\begin{aligned}
&\frac{1}{C^2}\sum_{j,j'\in\mathcal{B}_t,j'\neq j}\mathrm{Tr}\left(\mathbb{E}_t\left[\mathbf{U}_{t,m,s}^\top\mathbf{e}_{t,m,s}\mathbf{e}_{t,m,s}^\top\mathbf{E}_{t,m,s}\mathbf{z}_m\mathbf{z}_m^\top 1_{\mathcal{G}_{t,s}}\right]\times\right.\\
&\qquad\qquad\qquad\qquad\left.\mathbb{E}_t\left[\mathbf{z}_{m'}\mathbf{z}_{m'}^\top\mathbf{E}_{t,m',s}^\top\mathbf{e}_{t,m',s}\mathbf{e}_{t,m',s}^\top\mathbf{U}_{t,m',s}1_{\mathcal{G}_{t,s}}\right]\right)\\
&\leq\frac{(C-1)c\sigma_{1,*}\mu_z k}{Cd^2}\mathcal{L}(\mathbf{U}_t,\mathbf{V}_t)1_{\mathcal{G}_{t,0}}.
\end{aligned}
\tag{34}
$$

by arguing as in (30). Similarly, for the second term,

$$\text{Tr}\left(\mathbb{E}_t\left[\|\mathbf{z}_m\|_2^2\mathbf{U}_{t,m,s}^\top\mathbf{e}_{t,m,s}\mathbf{e}_{t,m,s}^\top\mathbf{E}_{t,m,s}\mathbf{z}_m\mathbf{z}_m^\top\mathbf{E}_{t,m,s}^\top\mathbf{e}_{t,m,s}\mathbf{e}_{t,m,s}^\top\mathbf{U}_{t,m,s}1_{\mathcal{G}_{t,s}}\right]\right)$$

$$= \frac{1}{dM}\sum_{m=1}^{M}\sum_{i=1}^{d}\|\mathbf{z}_m\|_2^2\mathbb{E}_t\left[\|\mathbf{U}_{t,m,s}^\top\mathbf{e}_i\|_2^2(\mathbf{e}_i^\top\mathbf{E}_{t,m,s}\mathbf{z}_m)^2 1_{\mathcal{G}_{t,s}}\right]$$

$$\leq \frac{\mu_U r\sigma_{1,*}}{d^2 M}\sum_{m=1}^{M}\sum_{i=1}^{d}\|\mathbf{z}_m\|_2^2\mathbb{E}_t[(\mathbf{e}_i^\top\mathbf{E}_{t,m,s}\mathbf{z}_m)^2 1_{\mathcal{G}_{t,s}}] \tag{35}$$

$$\leq \frac{kr\mu_z\mu_U\sigma_{1,*}}{d^2 M}\sum_{m=1}^{M}\mathbb{E}_t[\mathcal{L}_m(\mathbf{U}_{t,m,s},\mathbf{V}_{t,m,s})1_{\mathcal{G}_{t,s}}]$$

$$\leq \frac{kr\mu_z\mu_U\sigma_{1,*}}{d^2}\mathcal{L}(\mathbf{U}_t,\mathbf{V}_t)1_{\mathcal{G}_{t,0}} \tag{36}$$

where (35) follows from the incoherence of $\mathbf{U}_{t,m,s}$ with respect to the standard basis. Equation (36) implies

$$\frac{1}{C^2}\text{Tr}\left(\sum_{m\in\mathcal{B}_t}\mathbb{E}_t\left[\|\mathbf{z}_m\|_2^2\mathbf{U}_{t,m,s}^\top\mathbf{e}_{t,m,s}\mathbf{e}_{t,m,s}^\top\mathbf{E}_{t,m,s}\mathbf{z}_m\mathbf{z}_m^\top\mathbf{E}_{t,m,s}^\top\mathbf{e}_{t,m,s}\mathbf{e}_{t,m,s}^\top\mathbf{U}_{t,m,s}1_{\mathcal{G}_{t,s}}\right]\right)$$

$$\leq \frac{kr\mu_z\mu_U\sigma_{1,*}}{Cd^2}\mathcal{L}(\mathbf{U}_t,\mathbf{V}_t)1_{\mathcal{G}_{t,0}} \tag{37}$$

Combining (37), (34) and (32), and using $1_{\mathcal{G}_{t,s}} \leq 1_{\mathcal{G}_{t,0}}$ yields

$$\mathbb{E}_t[\|\mathbf{V}_{t+1} - \mathbf{V}_t\|_F^2 1_{\mathcal{G}_{t,\tau}}] \leq \frac{\eta^2\tau^2 kr}{d^2}(c + \mu_U)\mu_z\sigma_{1,*}\mathcal{L}(\mathbf{U}_t,\mathbf{V}_t)1_{\mathcal{G}_{t,0}} \tag{38}$$

as desired. $\qquad\square$

**Lemma C.8.** *Define*

$$a_{t,s} := \left\|\frac{d}{M}\sum_{m=1}^{M}\mathbb{E}_t\left[\nabla_\mathbf{U}\hat{\mathcal{L}}_m(\mathbf{U}_{t,m,s},\mathbf{V}_{t,m,s},\mathbf{e}_{t,m,s})1_{\mathcal{G}_{t,s}} - \nabla_\mathbf{U}\mathcal{L}(\mathbf{U}_t,\mathbf{V}_t)1_{\mathcal{G}_{t,s}}\right]\right\|_F, \text{ and}$$

$$b_{t,s} := \left\|\frac{d}{M}\sum_{m=1}^{M}\mathbb{E}_t\left[\nabla_\mathbf{V}\hat{\mathcal{L}}_m(\mathbf{U}_{t,m,s},\mathbf{V}_{t,m,s},\mathbf{e}_{t,m,s})1_{\mathcal{G}_{t,s}} - \nabla_\mathbf{V}\mathcal{L}(\mathbf{U}_t,\mathbf{V}_t)1_{\mathcal{G}_{t,s}}\right]\right\|_F 1_{\mathcal{G}_{t,s}}.$$

*Then for any $t,s$:*

$$a_{t,s+1} \leq a_{t,s} + 6\frac{\eta}{d}c^3\mu^{3/2}\sigma_{1,*}^{3/2}\sqrt{\mathcal{L}(\mathbf{U}_t,\mathbf{V}_t)}1_{\mathcal{G}_{t,s}}, \text{ and} \tag{39}$$

$$b_{t,s+1} \leq b_{t,s} + 6\frac{\eta}{d}c^3\mu^{3/2}\sigma_{1,*}^{3/2}\sqrt{\mathcal{L}(\mathbf{U}_t,\mathbf{V}_t)}1_{\mathcal{G}_{t,s}}. \tag{40}$$

*Proof.* Recall $\mathbf{U}_{t,m,s+1} = \mathbf{U}_{t,m,s} - \eta\hat{\mathbf{G}}_{t,m,s}$ and $\mathbf{V}_{t,m,s+1} = \mathbf{V}_{t,m,s} - \eta\hat{\mathbf{H}}_{t,m,s}$, where

$$\hat{\mathbf{G}}_{t,m,s} = \mathbf{e}_{t,m,s}\mathbf{e}_{t,m,s}^\top(\mathbf{U}_{t,m,s}\mathbf{V}_{t,m,s}^\top - \mathbf{M}_*)\mathbf{z}_m\mathbf{z}_m^\top\mathbf{V}_{t,m,s}, \text{ and} \tag{41}$$

$$\hat{\mathbf{H}}_{t,m,s} = \mathbf{z}_m\mathbf{z}_m^\top(\mathbf{U}_{t,m,s}\mathbf{V}_{t,m,s}^\top - \mathbf{M}_*)^\top\mathbf{e}_{t,m,s}\mathbf{e}_{t,m,s}^\top\mathbf{U}_{t,m,s} \tag{42}$$

First, using that $\mathbf{e}_{t,m,s+1}$ is independent of all prior samples for all $m$, and $1_{\mathcal{G}_{t,s+1}} \leq 1_{\mathcal{G}_{t,s}}$, we have

$$a_{t,s+1} = \left\|\frac{d}{M}\sum_{m=1}^{M}\mathbb{E}_t[(\mathbf{e}_{t,m,s}\mathbf{e}_{t,m,s}^\top(\mathbf{U}_{t,m,s}\mathbf{V}_{t,m,s}^\top - \mathbf{M}_*)\mathbf{z}_m\mathbf{z}_m^\top\mathbf{V}_{t,m,s}\right.$$

$$\left. - \frac{1}{d}(\mathbf{U}_t\mathbf{V}_t^\top - \mathbf{M}_*)\mathbf{Z}\mathbf{Z}^\top\mathbf{V}_t)1_{\mathcal{G}_{t,s}}\right\|_F$$

$$= \frac{1}{M}\left\|\sum_{m=1}^{M}\mathbb{E}_t[((\mathbf{U}_{t,m,s+1}\mathbf{V}_{t,m,s+1}^\top - \mathbf{M}_*)\mathbf{z}_m\mathbf{z}_m^\top\mathbf{V}_{t,m,s+1}\right.$$

$$\left. - (\mathbf{U}_t\mathbf{V}_t^\top - \mathbf{M}_*)\mathbf{Z}\mathbf{Z}^\top\mathbf{V}_t)1_{\mathcal{G}_{t,s}}\right\|_F. \tag{43}$$

561    Next, we use $1_{\mathcal{G}_{t,s+1}} \leq 1_{\mathcal{G}_{t,s}}$ and the triangle inequality to obtain

$$a_{t,s+1}$$

$$\leq \frac{1}{M}\left\| \sum_{m=1}^{M} \mathbb{E}_t[((\mathbf{U}_{t,m,s+1}\mathbf{V}_{t,m,s+1}^\top - \mathbf{M}_*)\mathbf{z}_m\mathbf{z}_m^\top\mathbf{V}_{t,m,s+1} - (\mathbf{U}_t\mathbf{V}_t^\top - \mathbf{M}_*)\mathbf{Z}\mathbf{Z}^\top\mathbf{V}_t)1_{\mathcal{G}_{t,s}}]\right\|_F$$

$$= \frac{1}{M}\left\| \sum_{m=1}^{M} \mathbb{E}_t[(((\mathbf{U}_{t,m,s} - \eta\hat{\mathbf{G}}_{t,m,s})(\mathbf{V}_{t,m,s} - \eta\hat{\mathbf{H}}_{t,m,s})^\top - \mathbf{M}_*)\mathbf{z}_m\mathbf{z}_m^\top(\mathbf{V}_{t,m,s} - \eta\hat{\mathbf{H}}_{t,m,s}) \right.$$

$$\left. - (\mathbf{U}_t\mathbf{V}_t^\top - \mathbf{M}_*)\mathbf{Z}\mathbf{Z}^\top\mathbf{V}_t)1_{\mathcal{G}_{t,s}}]\right\|_F$$

$$\leq a_{t,s} + \left\| \frac{\eta}{M}\sum_{m=1}^{M}\mathbb{E}_t[\mathbf{M}_*\mathbf{z}_m\mathbf{z}_m^\top\hat{\mathbf{H}}_{t,m,s}1_{\mathcal{G}_{t,s}}]\right\|_F + \left\| \frac{\eta}{M}\sum_{m=1}^{M}\mathbb{E}_t[\mathbf{U}_{t,m,s}\mathbf{V}_{t,m,s}^\top\mathbf{z}_m\mathbf{z}_m^\top\hat{\mathbf{H}}_{t,m,s}1_{\mathcal{G}_{t,s}}]\right\|_F$$

$$+ \left\| \frac{\eta^2}{M}\sum_{m=1}^{M}\mathbb{E}_t[\mathbf{U}_{t,m,s}\hat{\mathbf{H}}_{t,m,s}^\top\mathbf{z}_m\mathbf{z}_m^\top\hat{\mathbf{H}}_{t,m,s}1_{\mathcal{G}_{t,s}}]\right\|_F + \left\| \frac{\eta^2}{M}\sum_{m=1}^{M}\mathbb{E}_t[\hat{\mathbf{G}}_{t,m,s}\mathbf{V}_{t,m,s}^\top\mathbf{z}_m\mathbf{z}_m^\top\hat{\mathbf{H}}_{t,m,s}1_{\mathcal{G}_{t,s}}]\right\|_F$$

$$+ \left\| \frac{\eta^3}{M}\sum_{m=1}^{M}\mathbb{E}_t[\hat{\mathbf{G}}_{t,m,s}\hat{\mathbf{H}}_{t,m,s}^\top\mathbf{z}_m\mathbf{z}_m^\top\hat{\mathbf{H}}_{t,m,s}1_{\mathcal{G}_{t,s}}]\right\|_F + \left\| \frac{\eta}{M}\sum_{m=1}^{M}\mathbb{E}_t[\hat{\mathbf{G}}_{t,m,s}\mathbf{V}_{t,m,s}^\top\mathbf{z}_m\mathbf{z}_m^\top\mathbf{V}_{t,m,s}1_{\mathcal{G}_{t,s}}]\right\|_F$$

$$+ \left\| \frac{\eta^2}{M}\sum_{m=1}^{M}\mathbb{E}_t[\hat{\mathbf{G}}_{t,m,s}\hat{\mathbf{H}}_{t,m,s}^\top\mathbf{z}_m\mathbf{z}_m^\top\mathbf{V}_{t,m,s}1_{\mathcal{G}_{t,s}}]\right\|_F + \left\| \frac{\eta}{M}\sum_{m=1}^{M}\mathbb{E}_t[\mathbf{U}_{t,m,s}\hat{\mathbf{H}}_{t,m,s}^\top\mathbf{z}_m\mathbf{z}_m^\top\mathbf{V}_{t,m,s}1_{\mathcal{G}_{t,s}}]\right\|_F.$$

562    We first consider the terms that involve only one stochastic gradient. We have

$$\left\| \frac{\eta}{M}\sum_{m=1}^{M}\mathbb{E}_t[\mathbf{M}_*\mathbf{z}_m\mathbf{z}_m^\top\hat{\mathbf{H}}_{t,m,s}1_{\mathcal{G}_{t,s}}]\right\|_F$$

$$= \left\| \frac{\eta}{M}\sum_{m=1}^{M}\mathbb{E}_t[\mathbf{M}_*\mathbf{z}_m\mathbf{z}_m^\top\mathbf{z}_m\mathbf{z}_m^\top(\mathbf{U}_{t,m,s}\mathbf{V}_{t,m,s}^\top - \mathbf{M}_*)^\top\mathbf{e}_{t,m,s}\mathbf{e}_{t,m,s}^\top\mathbf{U}_{t,m,s}1_{\mathcal{G}_{t,s}}]\right\|_F$$

$$= \left\| \frac{\eta}{dM}\sum_{m=1}^{M}\|\mathbf{z}_m\|_2^2\mathbb{E}_t[\mathbf{M}_*\mathbf{z}_m\mathbf{z}_m^\top(\mathbf{U}_{t,m,s}\mathbf{V}_{t,m,s}^\top - \mathbf{M}_*)^\top\mathbf{U}_{t,m,s}1_{\mathcal{G}_{t,s}}]\right\|_F$$

$$\leq \frac{\eta}{dM}\sum_{m=1}^{M}\|\mathbf{z}_m\|_2^2\mathbb{E}_t[\|\mathbf{M}_*\mathbf{z}_m\mathbf{z}_m^\top(\mathbf{U}_{t,m,s}\mathbf{V}_{t,m,s}^\top - \mathbf{M}_*)^\top\mathbf{U}_{t,m,s}1_{\mathcal{G}_{t,s}}\|_F]$$

$$\leq \frac{\eta}{dM}\sum_{m=1}^{M}\|\mathbf{z}_m\|_2^2\|\mathbf{M}_*\mathbf{z}_m\|_2\mathbb{E}_t[\|\mathbf{z}_m^\top(\mathbf{U}_{t,m,s}\mathbf{V}_{t,m,s}^\top - \mathbf{M}_*)^\top\|\|\mathbf{U}_{t,m,s}\|_2 1_{\mathcal{G}_{t,s}}]$$

$$\leq \frac{\eta k^{3/2}}{d}c\mu_z^{3/2}\sigma_{1,*}^{3/2}\frac{1}{M}\sum_{m=1}^{M}\mathbb{E}_t\left[\sqrt{\mathcal{L}_m(\mathbf{U}_{t,m,s}, \mathbf{V}_{t,m,s})}1_{\mathcal{G}_{t,s}}\right]$$

$$\leq \frac{\eta k^{3/2}}{d}c\mu_z^{3/2}\sigma_{1,*}^{3/2}\sqrt{\mathcal{L}(\mathbf{U}_t, \mathbf{V}_t)}1_{\mathcal{G}_{t,0}}, \tag{44}$$

563    where the last inequality follows by definition of $1_{\mathcal{G}_{t,s}}$ and $1_{\mathcal{G}_{t,s}} \leq 1_{\mathcal{G}_{t,0}}$. We can similarly show that

$$\left\| \frac{\eta}{M}\sum_{m=1}^{M}\mathbb{E}_t[\mathbf{U}_{t,m,s}\hat{\mathbf{H}}_{t,m,s}^\top\mathbf{z}_m\mathbf{z}_m^\top\mathbf{V}_{t,m,s}1_{\mathcal{G}_{t,s}}]\right\|_F \leq \frac{\eta k^{3/2}}{d}c^3\mu^{3/2}\sigma_{1,*}^{3/2}\sqrt{\mathcal{L}(\mathbf{U}_t, \mathbf{V}_t)}1_{\mathcal{G}_{t,0}} \tag{45}$$

$$\left\| \frac{\eta}{M}\sum_{m=1}^{M}\mathbb{E}_t[\hat{\mathbf{G}}_{t,m,s}\mathbf{V}_{t,m,s}^\top\mathbf{z}_m\mathbf{z}_m^\top\mathbf{V}_{t,m,s}1_{\mathcal{G}_{t,s}}]\right\|_F \leq \frac{\eta k^{3/2}}{d}c^3\mu^{3/2}\sigma_{1,*}^{3/2}\sqrt{\mathcal{L}(\mathbf{U}_t, \mathbf{V}_t)}1_{\mathcal{G}_{t,0}}. \tag{46}$$

564 Next, we consider the terms that involve products of two stochastic gradients. Note that

$$\mathbb{E}_t[\hat{\mathbf{G}}_{t,m,s}\hat{\mathbf{H}}_{t,m,s}^\top \mathbf{z}_m\mathbf{z}_m^\top \mathbf{V}_{t,m,s}]$$
$$= \mathbb{E}_t[\mathbf{e}_{t,m,s}\mathbf{e}_{t,m,s}^\top \mathbf{E}_{t,m,s}\mathbf{z}_m\mathbf{z}_m^\top \mathbf{V}_{t,m,s}\mathbf{U}_{t,m,s}^\top \mathbf{e}_{t,m,s}\mathbf{e}_{t,m,s}^\top \mathbf{E}_{t,m,s}\mathbf{z}_m\mathbf{z}_m^\top \mathbf{z}_m\mathbf{z}_m^\top \mathbf{V}_{t,m,s}]$$
$$= \frac{\|\mathbf{z}_m\|_2^2}{d} \mathbb{E}_t\left[\mathrm{diag}([(\mathbf{e}_i^\top \mathbf{E}_{t,m,s}\mathbf{z}_m)(\mathbf{e}_i^\top \mathbf{U}_{t,m,s}\mathbf{V}_{t,m,s}^\top \mathbf{z}_m)]_{i\in[d]}) \, \mathbf{E}_{t,m,s}\mathbf{z}_m\mathbf{z}_m^\top \mathbf{V}_{t,m,s}\right] \qquad (47)$$

565 Next, the incoherence and norm boundedness conditions in the event $\mathcal{G}_{t,s}$ imply that for each $i \in [d]$,

$$|(\mathbf{e}_i^\top \mathbf{E}_{t,m,s}\mathbf{z}_m)(\mathbf{e}_i^\top \mathbf{U}_{t,m,s}\mathbf{V}_{t,m,s}^\top \mathbf{z}_m)|1_{\mathcal{G}_{t,s}} \le c^2\sigma_{1,*}\frac{\sqrt{kr\mu_E\mu_U\mu_z}}{d}\sqrt{\mathcal{L}_m(\mathbf{U}_{t,m,s}, \mathbf{V}_{t,m,s})}1_{\mathcal{G}_{t,s}}.$$

566 Therefore

$$\left\|\frac{\eta^2}{M}\sum_{m=1}^M \mathbb{E}_t[\hat{\mathbf{G}}_{t,m,s}\hat{\mathbf{H}}_{t,m,s}^\top \mathbf{z}_m\mathbf{z}_m^\top \mathbf{V}_{t,m,s}1_{\mathcal{G}_{t,s}}]\right\|_F \le c^3\frac{\eta^2k^2}{d^2}\sigma_{1,*}^{3/2}\mu_z^2\sqrt{r\mu_E\mu_U}\mathbb{E}_t[\mathcal{L}_m(\mathbf{U}_{t,m,s}, \mathbf{V}_{t,m,s})1_{\mathcal{G}_{t,s}}]$$
$$\le c^3\frac{\eta^2k^2}{d^2}\sigma_{1,*}^{3/2}\mu_z^2\sqrt{r\mu_E\mu_U}\mathcal{L}(\mathbf{U}_t, \mathbf{V}_t)1_{\mathcal{G}_{t,0}} \quad (48)$$

567 Similarly, for the other second-order terms we have

$$\left\|\frac{\eta^2}{M}\sum_{m=1}^M \mathbb{E}_t[\hat{\mathbf{G}}_{t,m,s}\mathbf{V}_{t,m,s}^\top \mathbf{z}_m\mathbf{z}_m^\top \hat{\mathbf{H}}_{t,m,s}]\right\|_F 1_{\mathcal{G}_{t,s}} \le c^3\frac{\eta^2k^2}{d^2}\sigma_{1,*}^{3/2}\mu_z^2\sqrt{r\mu_E\mu_U}\mathcal{L}(\mathbf{U}_t, \mathbf{V}_t)1_{\mathcal{G}_{t,0}}$$

568 and

$$\left\|\frac{\eta^2}{M}\sum_{m=1}^M \mathbb{E}_t[\mathbf{U}_{t,m,s}\hat{\mathbf{H}}_{t,m,s}^\top \mathbf{z}_m\mathbf{z}_m^\top \hat{\mathbf{H}}_{t,m,s}1_{\mathcal{G}_{t,s}}]\right\|_F$$
$$= \left\|\frac{\eta^2}{dM}\sum_{m=1}^M \|\mathbf{z}_m\|_2^4 \mathbf{U}_{t,m,s}\mathbf{U}_{t,m,s}^\top \mathrm{diag}([(\mathbf{e}_i^\top \mathbf{E}_{t,m,s}\mathbf{z}_m)^2]_{i\in[d]})\mathbf{U}_{t,m,s}\right\|_F 1_{\mathcal{G}_{t,s}}$$
$$\le \frac{\eta^2k^2}{d^2}c^3\sigma_{1,*}^{3/2}\sqrt{r}\mu_z^2\mu_E\mathbb{E}_t[\mathcal{L}_m(\mathbf{U}_{t,m,s}, \mathbf{V}_{t,m,s})1_{\mathcal{G}_{t,s}}]$$
$$\le \frac{\eta^2k^2}{d^2}c^3\sigma_{1,*}^{3/2}\sqrt{r}\mu_z^2\mu_E\mathcal{L}(\mathbf{U}_t, \mathbf{V}_t)1_{\mathcal{G}_{t,0}} \qquad (49)$$

569 For the third-order term, we have

$$\mathbb{E}_t[\hat{\mathbf{G}}_{t,m,s}\hat{\mathbf{H}}_{t,m,s}^\top \mathbf{z}_m\mathbf{z}_m^\top \hat{\mathbf{H}}_{t,m,s}]$$
$$= \|\mathbf{z}_m\|_2^4\mathbb{E}_t[(\mathbf{e}_{t,m,s}^\top \mathbf{E}_{t,m,s}\mathbf{z}_m)^3(\mathbf{z}_m^\top \mathbf{V}_{t,m,s}\mathbf{U}_{t,m,s}^\top \mathbf{e}_{t,m,s})\mathbf{e}_{t,m,s}\mathbf{e}_{t,m,s}^\top \mathbf{U}_{t,m,s}]$$
$$= \frac{\|\mathbf{z}_m\|_2^4}{d}\mathbb{E}_t[\mathrm{diag}([(\mathbf{e}_i^\top \mathbf{E}_{t,m,s}\mathbf{z}_m)^3(\mathbf{z}_m^\top \mathbf{V}_{t,m,s}\mathbf{U}_{t,m,s}^\top \mathbf{e}_{t,m,s})]_{i\in[d]})\mathbf{U}_{t,m,s}], \qquad (50)$$

570 and using the properties of $\mathcal{G}_{t,s}$, for each $i \in [d]$,

$$|(\mathbf{e}_i^\top \mathbf{E}_{t,m,s}\mathbf{z}_m)^3(\mathbf{z}_m^\top \mathbf{V}_{t,m,s}\mathbf{U}_{t,m,s}^\top \mathbf{e}_i)|1_{\mathcal{G}_{t,s}} \le \frac{c^2}{d^2}\mu_E^{3/2}\sqrt{kr\mu_z\mu_U}\sigma_{1,*}\mathcal{L}_m^{3/2}(\mathbf{U}_{t,m,s}, \mathbf{V}_{t,m,s})1_{\mathcal{G}_{t,s}}$$

571 This implies that

$$\left\|\frac{\eta^3}{M}\sum_{m=1}^M \mathbb{E}_t[\hat{\mathbf{G}}_{t,m,s}\hat{\mathbf{H}}_{t,m,s}^\top \mathbf{z}_m\mathbf{z}_m^\top \hat{\mathbf{H}}_{t,m,s}1_{\mathcal{G}_{t,s}}]\right\|_F$$
$$\le \frac{\eta^3k^{5/2}r}{d^3}c^3\mu_E^{3/2}\mu_z^{5/2}\sqrt{\mu_U}\sigma_{\max,*}^{3/2}\mathbb{E}_t[\mathcal{L}_m^{3/2}(\mathbf{U}_{t,m,s}, \mathbf{V}_{t,m,s})1_{\mathcal{G}_{t,s}}]$$
$$\le \frac{\eta^3k^{5/2}r}{d^3}c^3\mu_E^{3/2}\mu_z^{5/2}\sqrt{\mu_U}\sigma_{\max,*}^{3/2}\mathcal{L}^{3/2}(\mathbf{U}_t, \mathbf{V}_t)1_{\mathcal{G}_{t,0}}. \qquad (51)$$

Combining the bounds on all terms yields

$$a_{t,s+1} \leq a_{t,s} + 3\frac{\eta k^{3/2}}{d}c^3\mu^{3/2}\sigma_{1,*}^{3/2}\sqrt{\mathcal{L}(\mathbf{U}_t, \mathbf{V}_t)}1_{\mathcal{G}_{t,0}} + 3\frac{\eta^2 k^2\sqrt{r}}{d^2}c^3\sigma_{1,*}^{3/2}\mu^3\mathcal{L}(\mathbf{U}_t, \mathbf{V}_t)1_{\mathcal{G}_{t,0}}$$

$$+ \frac{\eta^3 k^{5/2}r}{d^3}c^3\mu^{9/2}\sigma_{1,*}^{3/2}\mathcal{L}^{3/2}(\mathbf{U}_t, \mathbf{V}_t)1_{\mathcal{G}_{t,0}}$$

$$\leq a_{t,s} + 6\frac{\eta k^{3/2}}{d}c^3\mu^{3/2}\sigma_{1,*}^{3/2}\sqrt{\mathcal{L}(\mathbf{U}_t, \mathbf{V}_t)}1_{\mathcal{G}_{t,0}} \tag{52}$$

using $\sqrt{\mathcal{L}(\mathbf{U}_t, \mathbf{V}_t)} \leq c_0\sigma_{r,*}$ and $\eta \leq \frac{c'\sigma_{r,*}}{\sqrt{kr}\mu^{3/2}}$.

The proof of (40) is analogous so we omit the details. □

**Lemma C.9.** *Let $a_{t,s}$ and $b_{t,s}$ be defined as in Lemma C.8. Then, for all $s = 0, \ldots, \tau$,*

$$\max(a_{t,s}, b_{t,s}) \leq 6\frac{\eta k^{3/2}\tau}{d}c^3\mu^{3/2}\sigma_{1,*}^{3/2}\sqrt{\mathcal{L}(\mathbf{U}_t, \mathbf{V}_t)}1_{\mathcal{G}_{t,0}}. \tag{53}$$

*Proof.* First note that since the average local gradient on the first local update is an unbiased estimate of the global gradient, we have $a_{t,0} = b_{t,0} = 0$. Applying Lemma C.8 recursively completes the proof. □

**Lemma C.10** (Bound on average client drift)**.**

$$\left\|\mathbb{E}_t\left[\frac{d}{M}\sum_{s=0}^{\tau-1}\sum_{m=1}^{M}\nabla\hat{\mathcal{L}}_m(\mathbf{U}_{t,m,s}, \mathbf{V}_{t,m,s}, \mathbf{e}_{t,m,s})1_{\mathcal{G}_{t,\tau}} - \tau\nabla\mathcal{L}(\mathbf{U}_t, \mathbf{V}_t)1_{\mathcal{G}_{t,\tau}}\right]\right\|_F$$

$$\leq 10\frac{\eta k^{3/2}\tau^2}{d}c^3\mu^{3/2}\sigma_{1,*}^{3/2}\sqrt{\mathcal{L}(\mathbf{U}_t, \mathbf{V}_t)}1_{\mathcal{G}_{t,0}}$$

*Proof.* We have

$$\left\|\mathbb{E}_t\left[\frac{d}{M}\sum_{s=0}^{\tau-1}\sum_{m=1}^{M}\nabla\hat{\mathcal{L}}_m(\mathbf{U}_{t,m,s}, \mathbf{V}_{t,m,s}, \mathbf{e}_{t,m,s})1_{\mathcal{G}_{t,\tau}} - \tau\nabla\mathcal{L}(\mathbf{U}_t, \mathbf{V}_t)1_{\mathcal{G}_{t,\tau}}\right]\right\|_F$$

$$\leq \sum_{s=0}^{\tau-1}\left\|\mathbb{E}_t\left[\frac{d}{M}\sum_{m=1}^{M}(\nabla\hat{\mathcal{L}}_m(\mathbf{U}_{t,m,s}, \mathbf{V}_{t,m,s}, \mathbf{e}_{t,m,s}) - \nabla\mathcal{L}(\mathbf{U}_t, \mathbf{V}_t))1_{\mathcal{G}_{t,s}}\right]\right\|_F$$

$$\leq \sum_{s=0}^{\tau-1}\left\|\frac{d}{M}\sum_{m=1}^{M}\mathbb{E}_t\left[\nabla\hat{\mathcal{L}}_m(\mathbf{U}_{t,m,s}, \mathbf{V}_{t,m,s}, \mathbf{e}_{t,m,s})1_{\mathcal{G}_{t,s}} - \nabla\mathcal{L}(\mathbf{U}_t, \mathbf{V}_t)1_{\mathcal{G}_{t,s}}\right]\right\|_F$$

$$= \sum_{s=1}^{\tau-1}\sqrt{a_{t,s}^2 + b_{t,s}^2} \tag{54}$$

$$\leq 10\frac{\eta k^{3/2}\tau^2}{d}c^3\mu^{3/2}\sigma_{1,*}^{3/2}\sqrt{\mathcal{L}(\mathbf{U}_t, \mathbf{V}_t)}1_{\mathcal{G}_{t,\tau}}. \tag{55}$$

where $a_{t,s}$ and $b_{t,s}$ are defined in Lemmas C.8, respectively, and (55) follows by Lemma C.9. □

**Lemma C.11.** *For any t,*

$$\mathbb{E}_t[\mathcal{L}(\mathbf{U}_{t+1}, \mathbf{V}_{t+1})1_{\mathcal{G}_{t,\tau}}] \leq \left(1 - 0.5\eta\tau\frac{\sigma_{r,*}}{cc_z^3 d}\right)\mathcal{L}(\mathbf{U}_t, \mathbf{V}_t)1_{\mathcal{G}_{t,0}}. \tag{56}$$

*Proof.* By Lemma C.6 and $1_{\mathcal{G}_{t,\tau}} \leq 1_{\mathcal{G}_{t,0}}$, there exists a particular $\beta > 0$ such that

$$\mathbb{E}_t[\mathcal{L}(\mathbf{U}_{t+1}, \mathbf{V}_{t+1})1_{\mathcal{G}_{t,\tau}}]$$
$$\leq \mathcal{L}(\mathbf{U}_t, \mathbf{V}_t)1_{\mathcal{G}_{t,0}} + \langle \nabla\mathcal{L}(\mathbf{U}_t, \mathbf{V}_t), \mathbb{E}_t[([\mathbf{U}_{t+1}; \mathbf{V}_{t+1}] - [\mathbf{U}_t; \mathbf{V}_t])1_{\mathcal{G}_{t,\tau}}]\rangle$$
$$+ \tfrac{\beta}{2}\mathbb{E}_t[\|\mathbf{U}_{t+1} - \mathbf{U}_t\|_F^2 1_{\mathcal{G}_{t,\tau}}] + \tfrac{\beta}{2}\mathbb{E}_t[\|\mathbf{V}_{t+1} - \mathbf{V}_t\|_F^2 1_{\mathcal{G}_{t,\tau}}] \tag{57}$$
$$= \mathcal{L}(\mathbf{U}_t, \mathbf{V}_t)1_{\mathcal{G}_{t,\tau}} + \langle \nabla\mathcal{L}(\mathbf{U}_t, \mathbf{V}_t), \frac{1}{M}\sum_{m=1}^{M}\mathbb{E}_t[([\mathbf{U}_{t,m,\tau}; \mathbf{V}_{t,m,\tau}] - [\mathbf{U}_t; \mathbf{V}_t])1_{\mathcal{G}_{t,\tau}}]\rangle$$
$$+ \tfrac{\beta}{2}\mathbb{E}_t[\|\mathbf{U}_{t+1} - \mathbf{U}_t\|_F^2 1_{\mathcal{G}_{t,\tau}}] + \tfrac{\beta}{2}\mathbb{E}_t[\|\mathbf{V}_{t+1} - \mathbf{V}_t\|_F^2 1_{\mathcal{G}_{t,\tau}}]$$
$$= \mathcal{L}(\mathbf{U}_t, \mathbf{V}_t)1_{\mathcal{G}_{t,\tau}} - \langle \nabla\mathcal{L}(\mathbf{U}_t, \mathbf{V}_t), \frac{\eta}{M}\sum_{m=1}^{M}\sum_{s=0}^{\tau-1}\mathbb{E}_t[\nabla\hat{\mathcal{L}}(\mathbf{U}_{t,m,s}, \mathbf{V}_{t,m,s})1_{\mathcal{G}_{t,\tau}}]\rangle$$
$$+ \tfrac{\beta}{2}\mathbb{E}_t[\|\mathbf{U}_{t+1} - \mathbf{U}_t\|_F^2 1_{\mathcal{G}_{t,\tau}}] + \tfrac{\beta}{2}\mathbb{E}_t[\|\mathbf{V}_{t+1} - \mathbf{V}_t\|_F^2 1_{\mathcal{G}_{t,\tau}}].$$

We now leverage that given $\mathcal{G}_{t,s}$, $\frac{d}{M}\sum_{m=1}^{M}\sum_{s=0}^{\tau-1}\mathbb{E}_t[\nabla\hat{\mathcal{L}}(\mathbf{U}_{t,m,s}, \mathbf{V}_{t,m,s})] \approx \tau\nabla\mathcal{L}(\mathbf{U}_t, \mathbf{V}_t)$, i.e. the average client gradient does not drift far from the global gradient, as shown in Lemma C.10. We also use that $\mathbb{E}_t[\|\mathbf{U}_{t+1} - \mathbf{U}_t\|_F^2 1_{\mathcal{G}_{t,\tau}}]$ and $\mathbb{E}_t[\|\mathbf{V}_{t+1} - \mathbf{V}_t\|_F^2 1_{\mathcal{G}_{t,\tau}}]$ are small by Lemma C.7, and $1_{\mathcal{G}_{t,\tau}} \leq 1_{\mathcal{G}_{t,0}}$.

$$\mathbb{E}_t[\mathcal{L}(\mathbf{U}_{t+1}, \mathbf{V}_{t+1})1_{\mathcal{G}_{t,\tau}}]$$
$$\leq \mathcal{L}(\mathbf{U}_t, \mathbf{V}_t)1_{\mathcal{G}_{t,0}} - \tfrac{\eta\tau}{d}\|\nabla\mathcal{L}(\mathbf{U}_t, \mathbf{V}_t)\|_F^2 1_{\mathcal{G}_{t,0}}$$
$$+ \tfrac{\eta}{d}\|\nabla\mathcal{L}(\mathbf{U}_t, \mathbf{V}_t)\|_F \sum_{s=0}^{\tau-1}\left\|\frac{d}{M}\sum_{m=1}^{M}\mathbb{E}_t\left[\nabla\hat{\mathcal{L}}_m(\mathbf{U}_{t,m,s}, \mathbf{V}_{t,m,s}, \mathbf{e}_{t,m,s})1_{\mathcal{G}_{t,s}} - \tau\nabla\mathcal{L}(\mathbf{U}_t, \mathbf{V}_t)1_{\mathcal{G}_{t,\tau}}\right]\right\|_F$$
$$+ \tfrac{\beta}{2}\mathbb{E}_t[\|\mathbf{U}_{t+1} - \mathbf{U}_t\|_F^2 1_{\mathcal{G}_{t,\tau}}] + \tfrac{\beta}{2}\mathbb{E}_t[\|\mathbf{V}_{t+1} - \mathbf{V}_t\|_F^2 1_{\mathcal{G}_{t,\tau}}]$$
$$\leq \mathcal{L}(\mathbf{U}_t, \mathbf{V}_t)1_{\mathcal{G}_{t,0}} - \tfrac{\eta\tau}{d}\|\nabla\mathcal{L}(\mathbf{U}_t, \mathbf{V}_t)\|_F^2 1_{\mathcal{G}_{t,0}}$$
$$+ 10\tfrac{\eta^2 k^{3/2}\tau^2}{d^2}c^3\mu^{3/2}\sigma_{1,*}^{3/2}\sqrt{\mathcal{L}(\mathbf{U}_t, \mathbf{V}_t)}\|\nabla\mathcal{L}(\mathbf{U}_t, \mathbf{V}_t)\|_F 1_{\mathcal{G}_{t,0}}$$
$$+ \tfrac{\beta^2}{2}\mathbb{E}_t[\|\mathbf{U}_{t+1} - \mathbf{U}_t\|_F^2 1_{\mathcal{G}_{t,\tau}}] + \tfrac{\beta^2}{2}\mathbb{E}_t[\|\mathbf{V}_{t+1} - \mathbf{V}_t\|_F^2 1_{\mathcal{G}_{t,\tau}}] \tag{58}$$
$$\leq (1 - \tfrac{\gamma\eta\tau}{d})\mathcal{L}(\mathbf{U}_t, \mathbf{V}_t)1_{\mathcal{G}_{t,0}} + 10\tfrac{\eta^2 k^{3/2}\tau^2}{d^2}c^3\mu^{3/2}\sigma_{1,*}^{3/2}\sqrt{\mathcal{L}(\mathbf{U}_t, \mathbf{V}_t)}\|\nabla\mathcal{L}(\mathbf{U}_t, \mathbf{V}_t)\|_F 1_{\mathcal{G}_{t,0}}$$
$$+ \tfrac{\beta}{2}\mathbb{E}_t[\|\mathbf{U}_{t+1} - \mathbf{U}_t\|_F^2 1_{\mathcal{G}_{t,\tau}}] + \tfrac{\beta}{2}\mathbb{E}_t[\|\mathbf{V}_{t+1} - \mathbf{V}_t\|_F^2 1_{\mathcal{G}_{t,\tau}}] \tag{59}$$
$$\leq (1 - \tfrac{\gamma\eta\tau}{d})\mathcal{L}(\mathbf{U}_t, \mathbf{V}_t)1_{\mathcal{G}_{t,0}} + 10\tfrac{\eta^2 k^{3/2}\tau^2}{d^2}c^4\sqrt{c_z}\mu^{3/2}\sigma_{1,*}^2\mathcal{L}(\mathbf{U}_t, \mathbf{V}_t)1_{\mathcal{G}_{t,0}}$$
$$+ \beta\tfrac{\eta^2 kr}{d}\tau^2 c\mu^2\sigma_{1,*}\mathcal{L}(\mathbf{U}_t, \mathbf{V}_t)1_{\mathcal{G}_{t,0}} \tag{60}$$

where (58) follows by Lemma C.9 and (59) follows by Lemma C.5, where $\gamma$ is defined therein, and (60) follows by Lemma C.7 and the fact that

$$\|\nabla\mathcal{L}(\mathbf{U}_t, \mathbf{V}_t)\|_F 1_{\mathcal{G}_{t,0}} = \tfrac{1}{M}\sqrt{\|\mathbf{E}_t\mathbf{Z}^\top\mathbf{Z}\mathbf{V}_t\|_F^2 + \|\mathbf{U}_t^\top\mathbf{E}_t\mathbf{Z}^\top\mathbf{Z}\|_F^2}\, 1_{\mathcal{G}_{t,0}}$$
$$\leq \sqrt{2}c\tfrac{\sqrt{\sigma_{1,*}}}{M}\|\mathbf{Z}\|_2\|\mathbf{E}_t\mathbf{Z}^\top\|_F 1_{\mathcal{G}_{t,0}}$$
$$\leq c\sqrt{c_z\sigma_{1,*}\mathcal{L}(\mathbf{U}_t, \mathbf{V}_t)}1_{\mathcal{G}_{t,0}}. \tag{61}$$

Plugging in the values of $\gamma$ and $\beta$ from Lemmas C.5 and C.6, respectively, yields

$$\mathbb{E}_t[\mathcal{L}(\mathbf{U}_{t+1}, \mathbf{V}_{t+1})1_{\mathcal{G}_{t,\tau}}] \leq \left(1 - \tfrac{\eta\tau\sigma_{r,*}}{dcc_z^3} + 10\tfrac{\eta^2 k^{3/2}\tau^2}{d^2}c^4\sqrt{c_z}\mu^{3/2}\sigma_{1,*}^2\right.$$
$$\left. + \tfrac{\eta^2\tau^2 kr}{d}cc_z\left(6c^2 + 2\right)\sigma_{1,*}^2\mu^2\right)\mathcal{L}(\mathbf{U}_t, \mathbf{V}_t)1_{\mathcal{G}_{t,0}}$$
$$\leq \left(1 - \eta\tau\tfrac{\sigma_{r,*}}{cc_z^3 d} + 66\tfrac{\eta^2 k^{3/2}r}{d}\tau^2 c^4 c_z\mu^2\sigma_{1,*}^2\mu^2\right)\mathcal{L}(\mathbf{U}_t, \mathbf{V}_t)1_{\mathcal{G}_{t,0}}$$
$$\leq \left(1 - 0.5\eta\tau\tfrac{\sigma_{r,*}}{cc_z^3 d}\right)\mathcal{L}(\mathbf{U}_t, \mathbf{V}_t)1_{\mathcal{G}_{t,0}} \tag{62}$$

where the last inequality follows by choice of $\eta \leq \frac{\sigma_{r,*}}{132k^{3/2}r\tau\mu^2\sigma_{1,*}^2 c^5 c_z^4}$. $\qquad\square$

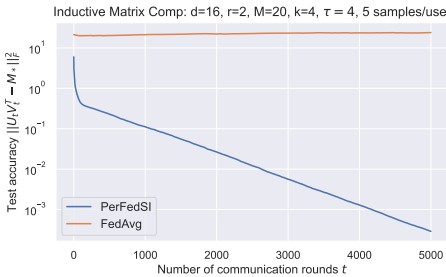

Figure 3: Federated matrix completion results on synthetic Gaussian data. Employing the side information (PerFedSI) leads to linear convergence to the ground-truth matrix, while not using the side information fails to converge to the ground-truth solution whatsoever.

We are finally ready to prove Theorem 3.3.

*Proof.* By Lemma C.11 and the fact that $\mathcal{G}_{t,\tau} \subset \mathcal{G}_{t-1,\tau}$, we have that for any $t$,

$$\mathbb{E}[\mathcal{L}(\mathbf{U}_t, \mathbf{V}_t)1_{\mathcal{G}_{t,\tau}}|\mathcal{S}_{t-1}] \leq \mathbb{E}[\mathcal{L}(\mathbf{U}_t, \mathbf{V}_t)1_{\mathcal{G}_{t-1,\tau}}|\mathcal{S}_{t-1}] \leq (1 - 0.5\eta\tau\frac{\sigma_{r,*}}{cc_z^3 d})\mathcal{L}(\mathbf{U}_{t-1}, \mathbf{V}_{t-1})1_{\mathcal{G}_{t,0}}$$
$$\leq (1 - 0.5\eta\tau\frac{\sigma_{r,*}}{cc_z^3 d})\mathcal{L}(\mathbf{U}_{t-1}, \mathbf{V}_{t-1})1_{\mathcal{G}_{t-1,\tau}} \tag{63}$$

Combining this with the Law of Total Expectation, we have

$$\mathbb{E}[\mathcal{L}(\mathbf{U}_T, \mathbf{V}_T)1_{\mathcal{G}_{T-1,\tau}}] = \mathbb{E}[\mathbb{E}[\mathcal{L}(\mathbf{U}_T, \mathbf{V}_T)1_{\mathcal{G}_{T-1,\tau}}|\mathcal{S}_{T-1}]]$$
$$\leq (1 - 0.5\eta\tau\frac{\sigma_{r,*}}{cc_z^3 d})\mathbb{E}[\mathcal{L}(\mathbf{U}_{T-1}, \mathbf{V}_{T-1})1_{\mathcal{G}_{T-1,\tau}}]$$
$$\leq (1 - 0.5\eta\tau\frac{\sigma_{r,*}}{cc_z^3 d})\mathbb{E}[\mathcal{L}(\mathbf{U}_{T-1}, \mathbf{V}_{T-1})1_{\mathcal{G}_{T-2,\tau}}] \tag{64}$$

where (64) follows by the fact that $1_{\mathcal{G}_{t,\tau}} \leq 1_{\mathcal{G}_{t-1,\tau}}$ for all $t$ since $\mathcal{G}_{t,\tau} \subset \mathcal{G}_{t-1,\tau}$. Applying (64) recursively completes the proof. $\qquad\square$

### C.4 Synthetic data simulations

Here verify our theoretical results by running an experiment on Gaussian data for the Inductive Matrix Completion problem. Here we sample ground-truth matrices $\mathbf{U}_*, \mathbf{V}_*$ and side information $\mathbf{Z}$ such that each element is an i.i.d. standard normal random variable. Then, $\mathbf{U}_*, \mathbf{V}_*$ are normalized via the QR factorization, and each row of $\mathbf{Z}$ is normalized. We use $d = 16, M = 20, k = 4, r = 2$. We then sample 5 indices per client that are the only indices observed for that client throughout the entire training process. We run FedAvg with $\tau$ local updates with and without side information, where each local update approximates the local gradient by sampling one of the pre-sampled 5 indices. The results are shown in Figure C.4. Using the side information to solve the dimension-reduced problem (PerFedSI) leads to linear convergence to the ground-truth solution, while solving the original problem (vanilla FedAvg) does not lead to convergence to the ground-truth.

## D Experiments

All experiments were run in PyTorch and used four-layer convolutional neural networks with convolutional layer batch normalization, ReLU activation, and 2x2 max pooling, followed by a final linear layer. All methods sample 20% of clients are sampled per round, use SGD with momentum parameter 0.5 and data batch size 10 for local updates. Grid search over {0.5, 0.1, 0.05, 0.01} was used to select the learning rates, and all methods use learning rate of 0.05 for Omniglot and 0.1 for the CIFAR experiments, besides Ditto which used learning rate 0.05 in all cases. Ditto also used regularization parameter $\mu = 1$ in all cases. Test accuracy was evaluated on the local models. SR-PH treats the first two convolutional layers as personalized (local) and the rest of the layers as shared across all clients (global). SR-PH treats the four convolutional layers as shared and the linear layer as personalized. Additional details regarding the datasets are as follows.

Table 2: Number of Omniglot training and testing samples per character per client for different numbers of total clients $M$.

| | $M$ | | |
|---|---|---|---|
| | 50 | 200 | 500 |
| $\nu_{tr}$ | 16 | 4 | 1 |
| $\nu_{ts}$ | 4 | 1 | 1 |

**Omniglot.** Note that there are 20 samples per character in the Omniglot dataset. These were partitioned to clients such that if a client was assigned alphabet $A$, then that client has $\nu_{tr}$ training samples and $\nu_{ts}$ test samples from every character in $A$, where $\nu_{tr}$ and $\nu_{ts}$ are functions of the total number of clients $M$ as specified in Table D.

The same network architecture described above was used to train the alphabet embedding (with the last layer mapping to $\mathbb{R}^{50}$, corresponding to the number of alphabets, as each alphabet is a class in this case, rather than $\mathbb{R}^{1623}$ in the standard FL setup wherein a character is a class). The embedding was taken as the 256-dimensional output of the final convolutional layer prior to the linear layer. The side information for each client is taken as the average alphabet embedding of their training samples, and it is incorporated into the network by passing it through a linear layer mapping to $\mathbb{R}^{576}$ followed by a convolutional block, then concatenating the output to the input to the final linear layer of the network.

**CIFAR-10, CIFAR-100.** The datasets are first partitioned i.i.d. among clients. Then, one of four affine shifts is applied to each client's data (one shift per client). The four affine shifts are as follows: (i) 90 degree clockwise rotation + 3 degree clockwise shear, (2) 180 degree clockwise rotation + 6 degree clockwise shear, (3) 270 degree clockwise rotation + 9 degree clockwise shear, and (4) no shift.

