# OpenReview forum: "PerFedSI: A Framework for Personalized Federated Learning with Side Information"
_NeurIPS.cc/2022/Workshop/Federated_Learning — FL-NeurIPS 2022 Poster_

### Official Review · Reviewer_srgV · 2022-10-17
**Interesting setting that needs stronger motivation in practice**

This paper proposed personalized federated learning with side information. For example, "side information" of clients that can be potentially represented as an embedding vector is combined with the feature vectors of samples for classification tasks. Two combination models, concatenation and element-wise multiplication, are considered. Experiments run on Omniglot where a pretrained alphabet classifier is used as  side information, and CIFAR-10/100 where affine transforms are used for both data heterogeneity and side information.

The setting of side information is generally interesting and intuitive, and I hope the following comments can help improve the paper
(1) The connection between theory and practice seems to be loose. While it is not uncommon to have a gap between theory and practice, the gap in this paper seems to be particularly large. I would strongly encourage the authors to provide explicit connections.
(2) While the story of side information seems to make sense, the experimental setting seems to be less convincing. The experiments assume a strong correlation between heterogeneity and the known side information. It is not quite clear how the side information can be acquired in practice.
(3) Some technical choices for incorporating side information seems to be ad-hoc. Some ablation study might help here, for example, for Omniglot, what is the performance if all clients use the the same side information?

---

### Official Review · Reviewer_fkev · 2022-10-18
**Interesting problem setting.**

This paper considers the federated learning problem with client-specific side information. In practice, such side information could be given to the service provider, and by leveraging it, FL algorithms could provide much better personalized performance than the case without the side information. This paper provides two possible approaches that can deal with the side information: concatenation and element-wise multiplication. The element-wise multiplication part is theoretically analyzed with a simple low rank matrix completion setting.

Pros.

This paper considers an interesting problem setting FL with side information

Cons.

Among two approaches, only the element-wise multiplication method is analyzed under a simple setting.

The simulation is performed only with synthetic settings.

---

### Official Review · Reviewer_1Zxk · 2022-10-19
**The paper proposes a novel approach for incorporating side information for improved personalization. The authors show theoretical as well as empirical analyses. However, the algorithm needs to be clear on its compatibility with secure aggregation as sharing side information may jeopardize privacy. Additionally, the paper lacks an overview of the performances of different methods of incorporating side information.**

The authors propose the use of auxiliary information about the clients’ data domain or models. The authors formulate the method to use this ‘side information’ in generic terms so as to encompass a wider variety of such methods. The authors deliver theoretical foundations showing the algorithm converges linearly as well as empirical results of the method on Omniglot, CIFAR-10, and CIFAR-100. Results show good performance compared to baselines.

Strengths:
+ The use of side information in the manner stated in the paper is new.
+ The paper proposes a general framework for using side information.
+ The paper gives a theoretical foundation for convergence
+ Empirical results show good performance

Weakness:
- It is not immediately clear if the client embeddings are compatible with secure aggregation. Sharing client information is against key principles of federated learning.
- The baseline comparison is not completely fair as the proposed method uses additional information. In the case of a small number of samples per client, having added information about client data distribution would definitely make any algorithm stand out. I think a comparison of different methods for incorporating side information would be more interpretable.
- The paper only shows empirical results for vision tasks only. The paper would benefit if the algorithm was applied in some other domains as well.

Questions:
- It seems the authors propose two methods of using side information but results are shown for only one. It is not immediately clear which one the results are for.

---

### Decision · Program_Chairs · 2022-10-20

Accept (Poster)